# Learning Parametric Distributions from Samples and Preferences

**Marc Jourdan** [1]   **Gizem Yüce** [1]   **Nicolas Flammarion** [1]

## Abstract

Recent advances in language modeling have underscored the role of preference feedback in enhancing model performance. This paper investigates the conditions under which preference feedback improves parameter estimation in classes of continuous parametric distributions. In our framework, the learner observes pairs of samples from an unknown distribution along with their relative preferences depending on the same unknown parameter. We show that preference-based M-estimators achieve a better asymptotic variance than sample-only M-estimators, further improved by deterministic preferences. Leveraging the hard constraints revealed by deterministic preferences, we propose an estimator achieving an estimation error scaling of $\mathcal{O}(1/n)$—a significant improvement over the $\Theta(1/\sqrt{n})$ rate attainable with samples alone. Next, we establish a lower bound that matches this accelerated rate; up to dimension and problem-dependent constants. While the assumptions underpinning our analysis are restrictive, they are satisfied by notable cases such as Gaussian or Laplace distributions for preferences based on the log-probability reward.

## 1. Introduction

Recent progress in language modeling has showcased the effectiveness of preference feedback for fine-tuning (Ziegler et al., 2019; Ouyang et al., 2022; Bai et al., 2022; Touvron et al., 2023; Dubey et al., 2024). Preference data—indicating relative quality between outcomes—consistently outperforms approaches using positive examples only like supervised fine-tuning (Ivison et al., 2024). This empirical success suggests that preference feedback introduces new, complementary information beyond the observed data. Understanding how and why preferences provide this advantage requires connecting the preference model to the

data-generating process (Ge et al., 2024).

To understand the role of preference feedback, we focus on a simpler yet illustrative problem: parameter estimation for parametric distributions and preferences. Specifically, the learner observes pairs of samples from an unknown distribution, along with preferences informed by the same parameter. For instance, preferences based on log-probabilities naturally link the preference and probability models, though other formulations are possible (Huang et al., 2024).

For continuous distributions, we uncover a significant statistical learning gap between preference-based and sample-only estimators. This paper primarily investigates this gap, taking the sample-only maximum likelihood estimator (MLE)—optimal among unbiased estimators—as a baseline. The well-established theory of M-estimators (Van der Vaart, 2000) suggests that preference-based M-estimators improve asymptotic variance under certain conditions. Yet, this improvement is modest: when samples are of similar quality, preference feedback approaches a fair coin toss, providing minimal additional information. While reducing asymptotic variance is encouraging, it does not fully explain the substantial performance gains observed empirically in large-scale language models.

For deterministic preferences, we prove a more striking result: preference-based estimators achieve a statistically significant acceleration in parameter estimation. Specifically, we show that the estimation error scales as $\mathcal{O}(1/n)$ instead of the $\mathcal{O}(1/\sqrt{n})$ rate achieved by sample-only estimators. This acceleration is supported by a matching lower bound, up to dimension and problem-dependent constants.

While this acceleration might sound surprising, the $\Theta(1/n)$ rate can already be observed in a special case of sample-only parameter estimation. For instance, consider estimating the location parameter $\theta$ of a uniform distribution on $[\theta, \theta + 1]$ based solely on samples (Wainwright, 2019). The minimax rate for estimation error is $\Theta(1/n)$. The optimal estimator achieving the accelerated rate is the minimum of uniform observations whose density is positive at $\theta$. This improved rate arises from the accumulation of random variables having a positive density at a specific point through a minimum (or maximum) operator, in contrast to the slower aggregation inherent to averaging. Similarly, for deterministic preferences with log-likelihood rewards, we observe

[1]School of Computer and Communication Sciences, EPFL, Lausanne, Vaud, Switzerland. Correspondence to: Marc Jourdan <marc.jourdan@epfl.ch>.

*Proceedings of the $42^{nd}$ International Conference on Machine Learning*, Vancouver, Canada. PMLR 267, 2025. Copyright 2025 by the author(s).

the true ordering between likelihoods. As it enforces hard constraints—through a minimum operator—on the admissible parameters, our preference-based estimator achieves accelerated convergence.

To illustrate this acceleration, consider the standard normal distribution with preferences based on log-probabilities. Let $n \in \mathbb{N}$ and $[n] := \{1, \cdots, n\}$. For each $i \in [n]$, observe samples $(X_i, Y_i) \sim \mathcal{N}(0_2, I_2)$ along with their log-likelihood deterministic preference $Z_i := \text{sign}((Y_i - X_i)S_i)$, where $S_i := (X_i + Y_i)/2$ is their average. The triplet $(X_i, Y_i, Z_i)$ imposes a hard constraint based on $S_i$ on the location of candidate estimators $\theta$ that are consistent with this log-likelihood deterministic preference. Specifically, they satisfy $\theta \leq S_i$ if $S_i > 0$, and $\theta \geq S_i$ otherwise. The set of feasible parameters satisfying all constraints is thus $\left[\max_{i \in [n], S_i < 0} S_i, \min_{i \in [n], S_i > 0} S_i\right]$. Since the density of $\mathcal{N}(0, 1/2)$ is positive near zero, the length of this interval decreases as $\mathcal{O}(1/n)$ with high probability.

## 1.1. Contributions

For continuous parametric probability distributions, we study the statistical learning gap between preference-based estimators and sample-only estimators.

- First, we show that preference-based M-estimators achieve a better asymptotic variance than sample-only M-estimators. The variance is further improved for deterministic preference.

- Second, we introduce an estimator satisfying the constraints revealed by the deterministic preferences, and prove an accelerated estimation error rate of $\mathcal{O}(1/n)$. This constitutes a significant improvement over the $\Theta(1/\sqrt{n})$ rate achieved by M-estimators.

- Third, we provide a lower bound of $\Omega(1/n)$, matching our upper bound up to problem-specific constants.

Our results are derived under general assumptions on the distributions and the preferences. While restrictive, they are satisfied by notable cases such as Gaussian or Laplace distributions for preferences based on log-probabilities.

## 1.2. Related Work

**Learning parametric distributions.** Parametric estimation is a central approach in statistics, reducing inference about a distribution to the estimation of a finite-dimensional parameter (Lehmann & Casella, 2006; Wasserman, 2013). The maximum likelihood estimator (MLE) is the most fundamental method in this setting. Its asymptotic properties are well studied (Cramér, 1946; Ibragimov & Has' Minskii, 2013; Van der Vaart, 2000), while non-asymptotic guarantees have been established in Birgé & Massart (1993) and Spokoiny (2012). Lower bounds in parametric estimation rely on techniques such as Le Cam's two-point

method (LeCam, 1973), Fano's method (Fano, 1952), and Assouad's method (Assouad, 1983), and provide fundamental limits on estimation accuracy (Tsybakov, 2009).

**Learning parametric value/preference functions.** In the tabular setting, learning from pairwise comparisons aligns with the ranking problem. The performance of MLE under the Bradley-Terry model (Bradley & Terry, 1952) and its extensions has been extensively studied (Hunter, 2004; Negahban et al., 2012; Hajek et al., 2014; Rajkumar & Agarwal, 2014; Shah et al., 2016; Shah & Wainwright, 2018; Mao et al., 2018). The continuous setting, where generalization beyond observed preferences is required, has received less attention, except for linear utility functions (Zhu et al., 2023; Ge et al., 2024; Yao et al., 2025). Beyond analyzing the sample complexity of reward learning with MLE under the Bradley-Terry noise model, Zhu et al. (2023) study the performance of policies trained on the learned reward model. They show that while MLE may fail, a pessimistic variant can yield a policy with improved performance. Relaxing the noise assumption, Ge et al. (2024) show that utility parameters remain unidentifiable without strong modeling assumptions, even with noise-free query responses. However, they demonstrate that, in the active learning setting, utility can still be learned, even in the absence of noise. Their results highlight that the sampling distribution of observations must be aligned with the utility function to achieve improved sample complexity. Yao et al. (2025) leverages sparsity in the preference model and establish sharp estimation rates depending on the sparsity level. Finally, related estimation problems have also been studied in the contexts of dueling bandits and reinforcement learning (Faury et al., 2020; Saha et al., 2023).

**Fine-tuning with preference data.** Large language models often go through a post-training phase focusing mainly on learning from preference feedback (Lambert, 2024), to improve capabilities such as summarization, instruction following, and reasoning. The standard approach, reinforcement learning from human feedback (RLHF) (Ziegler et al., 2019), trains a reward model to align with human preferences and then optimizes the policy using reinforcement learning, typically with PPO (Schulman et al., 2017). RLHF follows three main steps: supervised fine-tuning, reward model training, and policy optimization. Another line of work has explored alternatives to PPO to simplify training. One such method, direct preference optimization (DPO), reformulates the reward function to learn a policy directly from preference data, avoiding an explicit reward model. Other preference optimization objectives have also been proposed (Meng et al., 2024). Finally, while preference data has traditionally been gathered through human annotators, the learning paradigm has recently expanded to include self-play where the model critiques its own generations (Dubey

et al., 2024; Huang et al., 2024).

## 2. Problem Statement

**Parameter estimation.** Let $\Theta \subseteq \mathbb{R}^k$ be a set of parameters for a class of continuous probability distributions $\mathcal{F}$ over $\mathcal{X} \subseteq \mathbb{R}^d$. Let $B_\Theta := \max_{\theta \in \Theta} \|\theta\|$ be the bound on $\Theta$ for the norm $\|\cdot\|$ specific to $\mathcal{F}$. Let $\mathcal{S}_{k-1}$ be the unit sphere for this norm. Let $p_\theta^{\otimes 2}$ be the distribution of two independent observations of $p_\theta$.

Let $\theta^\star$ be an unknown parameter to estimate. Our samples are drawn from $p_{\theta^\star}^{\otimes 2}$, i.e., $(X, Y) \sim p_{\theta^\star}^{\otimes 2}$. We use two archetypal examples satisfying our assumptions. First, the class $\mathcal{F}_{\mathcal{N}, \Sigma}$ of multivariate Gaussian distributions with known covariance $\Sigma$, where $\Theta$ are the natural parameters with norm $\|\cdot\|_\Sigma$ where $\|x\|_\Sigma := \sqrt{x^\intercal \Sigma x}$. Second, the class $\mathcal{F}_{\mathrm{Lap}, b}$ of Laplace distributions with known scale $b$, where $\Theta$ are the mean parameters with norm $|\cdot|$.

**Preference feedback.** Let $\ell_\theta : \mathcal{X}^2 \to \mathbb{R}$ be a parametric preference function. Given a parametric reward function $r_\theta$, a reward-based preference function is defined as $\ell_\theta(x, y) = r_\theta(x) - r_\theta(y)$. As a concrete example for our derivations, we consider preference based on the log-probability reward $r_\theta = \log p_\theta$. Given observations $(x, y)$, the true preference $z$ of $x$ over $y$ is governed by $\mathrm{sign}(\ell_\theta(x, y)) \in \{\pm 1, 0\}$. In many settings, however, the observed preference $Z$ can be stochastic due to noise or randomness in human feedback.

Conditioned on $(X, Y) \sim p_\theta^{\otimes 2}$, we denote the p.d.f. of the law of the preference $Z$ by $h(\ell_\theta(X, Y), \cdot)$. On $\mathcal{X}^2 \times \{\pm 1, 0\}$, the p.d.f. of the law of $(X, Y, Z)$ is denoted as $q_{\theta, h}(x, y, z) := p_\theta^{\otimes 2}(x, y) h(\ell_\theta(x, y), z)$. Under deterministic feedback, the true preferences are observed:

$$h_{\mathrm{det}}(\cdot, z) := \mathbb{1}\left(z = \mathrm{sign}(\cdot)\right) . \tag{1}$$

Under stochastic feedback, noisy preferences $z \in \{\pm 1\}$ are observed based on the sigmoid link:

$$h_{\mathrm{sto}}(\cdot, z) := \sigma(z \cdot) \quad \text{with} \quad \sigma(x) := (1 + e^{-x})^{-1} . \tag{2}$$

**Informative preferences.** A natural question is to see when preference $Z \sim h(\ell_{\theta^\star}(X, Y), \cdot)$ helps to estimate $\theta^\star$ compared to using samples $(X, Y) \sim p_{\theta^\star}^{\otimes 2}$ only. Intuitively, given observations with null preference gradient, parameters close to $\theta^\star$ could have similar preferences. Therefore, those samples are not sufficient to discriminate between them. For that, let $\mathcal{G}_0(\theta^\star) = \{(x, y) \in \mathcal{X}^2 \mid |\ell_{\theta^\star}(x, y)| > 0\}$ (resp. $\mathcal{G}_1(\theta^\star) = \{(x, y) \in \mathcal{G}_0(\theta^\star) \mid \|\nabla_{\theta^\star} \ell_{\theta^\star}(x, y)\| > 0\}$) be the set of pairs with non-zero preference (resp. gradient) function. For observations in $\mathcal{G}_0(\theta^\star)^\complement$, the preference is zero, hence uninformative. For observations in $\mathcal{G}_1(\theta^\star)^\complement$, the preference is locally independent of the parameter. Therefore,

they do not provide gradient information to distinguish $\theta^\star$ from a neighboring alternative parameter. Only the preferences of samples in $\mathcal{G}_1(\theta^\star)$ can provide information on $\theta^\star$, hence preference learning is meaningful if these samples are observed, i.e., $\mathbb{P}_{p_{\theta^\star}^{\otimes 2}}(\mathcal{G}_1(\theta^\star)) > 0$ for all $\theta^\star \in \Theta$.

**Negative examples.** The above condition is restrictive both on $\ell_\theta$ and $p_\theta$, even when considering $r_\theta = \log p_\theta$. For example, taking $p_\theta$ as the uniform distribution over $[0, \theta]$, we have $\mathbb{P}_{p_\theta^{\otimes 2}}(\mathcal{G}_1(\theta)) = 0$.

### 2.1. Sample-only MLE

In the absence of preference observations, a natural baseline is to estimate $\theta^\star$ directly from the observations. Given $(X_i, Y_i)_{i \in [n]} \sim p_{\theta^\star}^{\otimes 2n}$, the sample-only (SO) MLE is

$$\widehat{\theta}_n^{\mathrm{SO}} \in \arg\min_\theta L_n^{\mathrm{SO}}(\theta) \quad \text{with}$$

$$L_n^{\mathrm{SO}}(\theta) := -\sum_{i \in [n]} \log p_\theta^{\otimes 2}(X_i, Y_i) . \quad \text{(SO MLE)}$$

**Asymptotic normality.** Under enough regularity (Van der Vaart, 2000), SO MLE is asymptotically normal, i.e.,

$$\sqrt{n}(\widehat{\theta}_n^{\mathrm{SO}} - \theta^\star) \rightsquigarrow_{n \to +\infty} \mathcal{N}(0_k, \mathcal{I}(p_{\theta^\star}^{\otimes 2})^{-1}) ,$$

where $\mathcal{I}(p_\theta) := \mathbb{E}_{p_\theta}[-\nabla_\theta^2 \log p_\theta]$ is the Fisher information matrix of $p_\theta$ and $\rightsquigarrow$ denote the convergence in distribution. Let $\succeq$ denote the Loewner order on p.s.d. matrices. By the Cramér-Rao bound (Rao, 1992), SO MLE has optimal asymptotic covariance among the class of unbiased sample-only estimators, i.e., all sample-only unbiased estimator with asymptotic variance $V$ satisfy $V \succeq \mathcal{I}(p_{\theta^\star}^{\otimes 2})^{-1}$.

While asymptotic guarantees provide insight into estimator behavior as $n \to \infty$, they do not capture performance in the relevant regime of moderate sample sizes. Modern statistics gives meaningful non-asymptotic concentration results on empirical estimators, e.g., for high-dimensional statistics (Vershynin, 2018; Wainwright, 2019).

**Regularity conditions.** The asymptotic statistics literature has devised weak regularity conditions under which asymptotic normality holds. "Classical conditions" assume stronger conditions, e.g., $\theta \mapsto \log p_\theta(x)$ is three times continuously differentiable for every $x \in \mathcal{X}$ and the integral of its third derivative converges uniformly for all $\theta$ (Van der Vaart, 2000, Chapter 5.6). Those "weak" or "classical" conditions ensure that integrals and derivatives can be exchanged, and Taylor approximations around $\theta^\star$ are well controlled. Throughout this paper, we use "under enough regularity" to refer to these regularity conditions on both $p_\theta$ and $\ell_\theta$.

For preferences based on the reward $r_\theta = \log p_\theta$, the regularity of $p_\theta$ implies the one of the preference $\ell_\theta$ due to the

properties of the logarithm. Moreover, those regularity conditions are satisfied for numerous well-known distributions such as $\mathcal{F}_{\mathcal{N},\Sigma}$ and $\mathcal{F}_{\mathrm{Lap},b}$. When studying deterministic preferences, we introduce general geometric assumptions on $p_\theta$ and $\ell_\theta$. Since these conditions are inherently more restrictive, our goal is not to identify the weakest possible regularity assumptions under which our derivations hold.

## 3. Preference-based M-estimator

In this section, we investigate when preference-based estimators can improve upon sample-only estimators. Given preference-labeled observations $\{(X_i, Y_i, Z_i)\}_{i\in[n]}$, we define the stochastic preferences MLE (SP MLE) as

$$\widehat{\theta}_n^{\mathrm{SP}} \in \arg\min_\theta L_n^{\mathrm{SP}}(\theta) \quad \text{with}$$

$$L_n^{\mathrm{SP}}(\theta) := L_n^{\mathrm{SO}}(\theta) - \sum_{i\in[n]} \log \sigma(Z_i \ell_\theta(X_i, Y_i)) . \quad \text{(SP MLE)}$$

This objective extends SO MLE by adding a preference-based term: a binary classification loss using the logistic function $(-\log\sigma(x))$. When preferences are stochastic, this estimator corresponds to the MLE under a probabilistic preference model, justifying its name. Under sufficient regularity, M-estimators achieve asymptotic normality, so our goal is to obtain lower asymptotic covariance for SP MLE than for SO MLE. In addition, we want to show that SP MLE reaches a lower asymptotic covariance for deterministic preferences than for stochastic preferences.

### 3.1. Stochastic Preferences

Under stochastic feedback, we are given noisy preference observations $(X_i, Y_i, Z_i)_{i\in[n]} \sim q_{\theta^\star, h_{\mathrm{sto}}}^{\otimes n}$, where $h_{\mathrm{sto}}$ is defined in Equation (2). SP MLE is a specific instance of M-estimator. Under enough regularity (Van der Vaart, 2000, Chapter 5.5), SP MLE is asymptotically normal, i.e.,

$$\sqrt{n}(\widehat{\theta}_n^{\mathrm{SP}} - \theta^\star) \rightsquigarrow_{n\to+\infty} \mathcal{N}(0_k, \mathcal{I}(q_{\theta, h_{\mathrm{sto}}})^{-1}) ,$$

where $\mathcal{I}(q_{\theta, h_{\mathrm{sto}}}) := \mathbb{E}_{q_{\theta, h_{\mathrm{sto}}}}[-\nabla_\theta^2 \log q_{\theta, h_{\mathrm{sto}}}]$ denotes the Fisher information matrix of $q_{\theta, h_{\mathrm{sto}}}$. By the Cramér-Rao bound (Rao, 1992), this variance is optimal among unbiased estimators that rely on stochastic preferences. Lemma 3.1 compares its efficiency to the sample-only MLE.

**Lemma 3.1.** *Let* $\Delta_\theta^{\mathrm{SP}} := \mathbb{E}_{p_{\theta^\star}^{\otimes 2}}[\sigma(\ell_\theta)\sigma(-\ell_\theta)\nabla_\theta\ell_\theta\nabla_\theta\ell_\theta^\intercal]$. *Then,* $\mathcal{I}(q_{\theta^\star, h_{\mathrm{sto}}}) = \mathcal{I}(p_{\theta^\star}^{\otimes 2}) + \Delta_{\theta^\star}^{\mathrm{SP}}$. *The p.s.d. matrix* $\Delta_{\theta^\star}^{\mathrm{SP}}$ *is definite if* $\mathbb{P}_{p_{\theta^\star}^{\otimes 2}}(|\langle u, \nabla_{\theta^\star}\ell_{\theta^\star}\rangle| > 0) > 0$ *for all* $u \in \mathcal{S}_{k-1}$.

Lemma 3.1 shows that $\mathcal{I}(q_{\theta^\star, h_{\mathrm{sto}}}) \succeq \mathcal{I}(p_{\theta^\star}^{\otimes 2})$ and exhibits a condition under which $\widehat{\theta}_n^{\mathrm{SP}}$ is asymptotically better than $\widehat{\theta}_n^{\mathrm{SO}}$, meaning that incorporating preference data improves asymptotic efficiency. The condition in Lemma 3.1 ensures that $\nabla_{\theta^\star}\ell_{\theta^\star}$ spans all directions with some probability, making the preference-based estimator asymptotically superior to the sample-only MLE.

For preferences based on the reward $r_\theta = \log p_\theta$, this condition holds for both Laplace and Gaussian distributions: $\Delta_{\theta^\star}^{\mathrm{SP}} = \frac{4}{b^2}\Delta_{\mathrm{Lap}(0,1)}^{\mathrm{SP}}$ for $\mathcal{F}_{\mathrm{Lap},b}$ (Appendix G), and $\Delta_{\theta^\star}^{\mathrm{SP}} = 2\Sigma^{1/2}\Delta_{\mathcal{N}(0_d, I_d)}^{\mathrm{SP}}\Sigma^{1/2}$ for $\mathcal{F}_{\mathcal{N},\Sigma}$ (Appendix F).

Thus, stochastic preferences can improve parameter estimation compared to sample-only estimators. However, non-asymptotic performance can differ, and in practice, the reduction in asymptotic variance may be small, as we investigate empirically in Section 6. Next, we examine whether M-estimators based on deterministic preferences can further improve upon their stochastic counterparts.

### 3.2. Deterministic Preferences

We now consider the setting where true preferences are observed, meaning that the preference labels $Z_i$ are deterministic. We observe $(X_i, Y_i, Z_i)_{i\in[n]} \sim q_{\theta^\star, h_{\mathrm{det}}}^{\otimes[n]}$, where $h_{\mathrm{det}}$ is defined in Equation (1). We use the same M-estimator as in the stochastic setting, $\widehat{\theta}_n^{\mathrm{SP}}(\theta) \in \arg\min_\theta L_n^{\mathrm{SP}}(\theta)$, but now with deterministic preferences. To distinguish this setting, we introduce the notation $\mathrm{SP}_{\mathrm{det}}$ for the preference-based estimator under deterministic feedback.

**Consistency of $\mathrm{SP}_{\mathrm{det}}$.** Define the population-level objective: $M(\theta) := \mathbb{E}_{p_{\theta^\star}^{\otimes 2}}[\log q_{\theta, h_{\mathrm{sto}}}(X, Y, \mathrm{sign}(\ell_{\theta^\star}(X, Y)))]$. Under enough regularity, $\widehat{\theta}_n^{\mathrm{SP}_{\mathrm{det}}}$ converges to a maximizer of $M(\theta)$ (Van der Vaart, 2000, Chapter 5.2). However, unlike in the stochastic setting, $\theta^\star$ may not be a maximizer of $M$ since standard regularity conditions on $p_\theta$ and $\ell_\theta$ are insufficient. A sufficient condition for consistency is

$$\mathbb{E}_{p_{\theta^\star}^{\otimes 2}}[\mathrm{sign}(\ell_{\theta^\star})\sigma(-|\ell_{\theta^\star}|)\nabla_{\theta^\star}\ell_{\theta^\star}] = 0_k , \quad (3)$$

which holds for $\mathcal{F}_{\mathcal{N},\Sigma}$ (Appendix F) and $\mathcal{F}_{\mathrm{Lap},b}$ (Appendix G) when using reward $r_\theta = \log p_\theta$.

**Asymptotic variance of $\mathrm{SP}_{\mathrm{det}}$.** If Equation (3) holds, then under additional regularity conditions (Van der Vaart, 2000, Chapter 5.3) $\mathrm{SP}_{\mathrm{det}}$ is asymptotically normal with covariance $V_{\theta^\star}^{\mathrm{SP}_{\mathrm{det}}}$ given by the following lemma.

**Lemma 3.2.** *Let* $H_{\theta^\star}^{\mathrm{SP}_{\mathrm{det}}} := \mathbb{E}_{p_{\theta^\star}^{\otimes 2}}\left[u_{\theta^\star}\nabla_{\theta^\star}^2\ell_{\theta^\star}\right]$, $\Delta_{\theta^\star}^{\mathrm{SP}_{\mathrm{det}}} := \mathbb{E}_{p_{\theta^\star}^{\otimes 2}}[(2\sigma(|\ell_{\theta^\star}|) - 1)\sigma(-|\ell_{\theta^\star}|)\nabla_{\theta^\star}\ell_{\theta^\star}\nabla_{\theta^\star}\ell_{\theta^\star}^\intercal]$ *and* $R_{\theta^\star}^{\mathrm{SP}_{\mathrm{det}}} := \mathbb{E}_{p_{\theta^\star}^{\otimes 2}}\left[u_{\theta^\star}(M_{\theta^\star} + M_{\theta^\star}^\intercal)\right]$ *where* $u_{\theta^\star} := \mathrm{sign}(\ell_{\theta^\star})\sigma(-|\ell_{\theta^\star}|)$ *and* $M_{\theta^\star} := -\nabla_{\theta^\star}\log p_{\theta^\star}^{\otimes 2}(\nabla_{\theta^\star}\ell_{\theta^\star})^\intercal$. *Then, we have* $V_{\theta^\star}^{\mathrm{SP}_{\mathrm{det}}} := V_{1,\theta^\star}^{-1}V_{2,\theta^\star}V_{1,\theta^\star}^{-1}$ *where* $V_{1,\theta^\star} = \mathcal{I}(q_{\theta^\star, h_{\mathrm{sto}}}) - H_{\theta^\star}^{\mathrm{SP}_{\mathrm{det}}}$ *and* $V_{2,\theta^\star} = \mathcal{I}(q_{\theta^\star, h_{\mathrm{sto}}}) - \Delta_{\theta^\star}^{\mathrm{SP}_{\mathrm{det}}} - R_{\theta^\star}^{\mathrm{SP}_{\mathrm{det}}}$. *If* $\mathbb{P}_{p_{\theta^\star}^{\otimes 2}}(|\ell_{\theta^\star}\langle u, \nabla_{\theta^\star}\ell_{\theta^\star}\rangle| > 0) > 0$ *for all* $u \in \mathcal{S}_{k-1}$, *the p.s.d. matrix* $\Delta_{\theta^\star}^{\mathrm{SP}_{\mathrm{det}}}$ *is definite.*

Equation (3) and $V_{\theta^\star}^{\mathrm{SP}_{\mathrm{det}}} \prec \mathcal{I}(q_{\theta^\star, h_{\mathrm{sto}}})^{-1}$ depend on the geometry of $p_{\theta^\star}^{\otimes 2}$ and $\ell_{\theta^\star}$. We verify that these conditions hold for $\mathcal{F}_{\mathcal{N},\Sigma}$ (Appendix F) and $\mathcal{F}_{\mathrm{Lap},b}$ (Appendix G) when

using reward $r_\theta = \log p_\theta$. For Laplace distribution, we have $H_{\theta^\star}^{\mathrm{SP_{det}}} = R_{\theta^\star}^{\mathrm{SP_{det}}} = 0$ and $\Delta_{\theta^\star}^{\mathrm{SP_{det}}} = \frac{4}{b^2} \Delta_{\mathrm{Lap}(0,1)}^{\mathrm{SP_{det}}}$. For Gaussian distributions, we have $H_{\theta^\star}^{\mathrm{SP_{det}}} = 0_{d \times d}$, $\Delta_{\theta^\star}^{\mathrm{SP_{det}}} = 2\Sigma^{1/2} \Delta_{\mathcal{N}(0_d, I_d)}^{\mathrm{SP_{det}}} \Sigma^{1/2}$ and $R_{\theta^\star}^{\mathrm{SP_{det}}} = 2\Sigma^{1/2} R_{\mathcal{N}(0_d, I_d)}^{\mathrm{SP_{det}}} \Sigma^{1/2}$ with $R_{\mathcal{N}(0_d, I_d)}^{\mathrm{SP_{det}}} \succeq 0_{d \times d}$.

Thus, deterministic preferences improve parameter estimation compared to stochastic preferences.

In conclusion, preference-based M-estimators provide asymptotic improvements in estimation efficiency. Next, we explore whether estimators beyond the M-estimation framework can achieve further gains, potentially exceeding the asymptotic normality limitations.

# 4. Beyond M-estimators

While computationally efficient, the $\mathrm{SP_{det}}$ estimator does not fully leverage the constraints imposed by deterministic preferences. Unlike in the stochastic setting, deterministic preferences provide separability: there exist parameters that classify training examples perfectly, including $\theta^\star$ itself. A key limitation of $\mathrm{SP_{det}}$ is that, like standard logistic regression, it minimizes a convex surrogate loss (negative log-likelihood). This approach can lead to misclassification of training examples.[1] This limitation suggests an opportunity to directly minimize the 0-1 loss[2], potentially achieving faster rates of convergence.

**0-1 loss minimization.** Given $(X_i, Y_i, Z_i)_{i \in [n]} \sim q_{\theta^\star, h_{\mathrm{det}}}^{\otimes [n]}$, we consider the set $\mathcal{C}_n$ of parameters that minimize the empirical $0 - 1$ loss, i.e.,

$$\mathcal{C}_n := \arg\min_{\theta \in \Theta} \sum_{i \in [n]} \mathbb{1}\left(Z_i \ell_\theta(X_i, Y_i) < 0\right) \qquad (4)$$
$$= \{\theta \in \Theta \mid \forall i \in [n], \ Z_i \ell_\theta(X_i, Y_i) \geq 0\} \ ,$$

which is non-empty as $\theta^\star \in \mathcal{C}_n$. Parameters $\theta \in \mathcal{C}_n$ perfectly classify all training examples. Any estimator $\widehat{\theta}_n^{\mathrm{AE}} \in \mathcal{C}_n$ is referred to as an arbitrary estimator (AE).

Alternatively, we constrain MLE to this feasible set, defining the deterministic preferences MLE (DP MLE), i.e.,

$$\widehat{\theta}_n^{\mathrm{DP}} \in \arg\min\{L_n^{\mathrm{SO}}(\theta) \mid \theta \in \mathcal{C}_n\} \qquad \text{(DP MLE)}$$

if $\widehat{\theta}_n^{\mathrm{SO}} \notin \mathcal{C}_n$, and $\widehat{\theta}_n^{\mathrm{DP}} := \widehat{\theta}_n^{\mathrm{SO}}$ otherwise. This estimator minimizes the negative log-likelihood of the samples while ensuring perfect preference classification. For Gaussian with $r_\theta = \log p_\theta$, $\widehat{\theta}_n^{\mathrm{DP}}$ estimates $\theta^\star$ better than $\widehat{\theta}_n^{\mathrm{SO}}$ for all $n$, i.e., DP MLE dominates SO MLE statistically.

---

[1]For binary classification with separable data, logistic regression converges in direction toward a separating hyperplane.

[2]For non-separable data, the minimization of the 0-1 classification loss can be NP-hard even for the simple class of linear classifiers, e.g., Feldman et al. (2012).

**Lemma 4.1.** *For all $n \in \mathbb{N}$ and almost surely, we have,*

$$\text{for } \mathcal{F}_{\mathcal{N}, \Sigma}, \quad \|\widehat{\theta}_n^{\mathrm{DP}} - \theta^\star\|_\Sigma \leq \|\widehat{\theta}_n^{\mathrm{SO}} - \theta^\star\|_\Sigma \ .$$

For stochastic preferences, minimizing the $0 - 1$ loss is generally NP-hard, requiring a convex surrogate like the logistic function. However, for deterministic preferences, computing $\mathcal{C}_n$ is more tractable. If $\theta \mapsto \ell_\theta$ is affine, then $\mathcal{C}_n$ is a convex polytope, defined by at most $n$ half-space constraints. For Gaussian-based preferences, i.e., $r_\theta = \log p_\theta$ and $\mathcal{F}_{\mathcal{N}, \Sigma}$, we have $Z_i \ell_\theta(X_i, Y_i) \geq 0$ if and only if $Z_i \langle X_i - Y_i, \theta - \Sigma^{-1}(X_i + Y_i)/2 \rangle \geq 0$.

**Consistency of $0 - 1$ loss minimization.** Define the disagreement probability between $\theta$ and $\theta^\star$ as $m(\theta) := \mathbb{P}_{p_{\theta^\star}^{\otimes 2}}(\mathcal{D}(\theta^\star, \theta))$ where $\mathcal{D}(\theta^\star, \theta) := \{(x, y) \in \mathcal{X}^2 \mid \ell_{\theta^\star}(x, y)\ell_\theta(x, y) < 0\}$ is the set of observations where $\theta$ and $\theta^\star$ assign informative yet opposite preferences.

Under enough regularity (Van der Vaart, 2000, Chapter 5.2), $\widehat{\theta}_n^{\mathrm{AE}}$ and $\widehat{\theta}_n^{\mathrm{DP}}$ converge in $\mathcal{C}(\theta^\star) := \{\theta \in \Theta \mid m(\theta) = 0\}$, which is the non-empty set of minimizers of $m(\theta)$ as $m(\theta) \geq 0 = m(\theta^\star)$. We note the set $\mathcal{C}(\theta^\star)$ contains $\theta^\star$, but possibly others. To ensure consistency ($\widehat{\theta}_n^{\mathrm{AE}}, \widehat{\theta}_n^{\mathrm{DP}} \to \theta^\star$), we impose the following identifiability assumption that guarantees $\mathcal{C}(\theta^\star) = \{\theta^\star\}$.

**Assumption 4.2** (Identifiability). *For all $\theta \neq \theta^\star$, $m(\theta) > 0$.*

When $r_\theta = \log p_\theta$, it holds for both Gaussian ($\mathcal{F}_{\mathcal{N}, \Sigma}$) (Appendix F) and Laplace ($\mathcal{F}_{\mathrm{Lap}, b}$) (Appendix G) cases.

**Fast estimation rate.** Once consistency is established, the next goal is to analyze the convergence rate of the estimation errors $\|\widehat{\theta}_n^{\mathrm{AE}} - \theta^\star\|$ and $\|\widehat{\theta}_n^{\mathrm{DP}} - \theta^\star\|$. Since these are not M-estimators, they are not necessarily limited to the typical parametric rate $\Omega(1/\sqrt{n})$.

Theorem 4.3 states our main result for Laplace and Gaussian distributions when using log-probability rewards, i.e., a high-probability accelerated rate in $\mathcal{O}(1/n)$.

**Theorem 4.3.** *Let $\delta \in (0, 1)$. For $\mathcal{F}_{\mathrm{Lap}, 1}$ and $\mathcal{F}_{\mathcal{N}, 1}$, we have, for all $n \geq \mathcal{O}(\log(1/\delta))$, with probability $1 - \delta$,*

$$\forall \widehat{\theta}_n \in \mathcal{C}_n, \quad n|\widehat{\theta}_n - \theta^\star| = \mathcal{O}\left(\log(1/\delta)\right) \ .$$

*For $\mathcal{F}_{\mathcal{N}, \Sigma}$ with $d > 1$, there exists positive $A_d =_{d \to +\infty} \mathcal{O}(\sqrt{d})$ such that, for all $n \geq \widetilde{\mathcal{O}}(\log(1/\delta))$, with probability $1 - \delta$,*

$$\forall \widehat{\theta}_n \in \mathcal{C}_n, \quad n\|\widehat{\theta}_n - \theta^\star\|_\Sigma \leq \mathcal{O}\left(A_d \log(1/\delta) \log n\right) \ .$$

Theorem 4.3 is a direct corollary of our main result, showing that $\max_{\theta \in \mathcal{C}_n} \|\theta - \theta^\star\| = \mathcal{O}(1/n)$ (see Theorem 4.8 below). It directly guarantees faster convergence rates for both $\widehat{\theta}_n^{\mathrm{AE}}$ and $\widehat{\theta}_n^{\mathrm{DP}}$. Theorem 4.8 holds under general geometric conditions on $p_\theta$ and $\ell_\theta$ that we introduce with intuitions, while sketching the proof in Section 4.1.

**Negative examples.** Assumption 4.2 is restrictive both on $\ell_\theta$ and $p_\theta$, even when considering $r_\theta = \log p_\theta$. For example, when all the distributions in $\mathcal{F}$ agree on their preferences, $\text{sign}(\ell_\theta(x, y))$ is independent of $\theta$. Therefore, we have $m(\theta) = 0$ for all $\theta \neq \theta^\star$, since $\ell_{\theta^\star}(x, y)\ell_\theta(x, y) \geq 0$. Such cases include scenarios where $p_\theta(x)$ is a monotonic function, e.g., the exponential distribution and the Pareto distribution with a known location, as well as the Laplace distribution with a known location. This motivates later assumptions on the directionality of $\nabla_{\theta^\star}\ell_{\theta^\star}$ for observed samples.

**Link to iterative human preference alignment.** Many human preference alignment methods build on the Bradley-Terry model for preference, based on rewards. Direct alignment algorithms use variants of the log-likelihood to define the implicit reward of a policy (Rafailov et al., 2023). Choosing $\ell_\theta(x, y) = \log p_\theta(x) - \log p_\theta(y)$ coincides with the optimal policy for maximum entropy RL (Swamy et al., 2025). When leveraging offline preference data, the assumption $(X, Y) \sim p_{\theta^\star}^{\otimes 2}$ is unrealistic, as $\ell_{\theta^\star}$ is collected from a fixed data set of pairs of observations. However, "online" preference data has become a popular paradigm in the training of recent LLMs. Those iterative alignment procedures rely on the preference data from an earlier model (Dubey et al., 2024). At stage $N$, the model $p_{\theta_N}$ is trained based on the preference data for generations by the previous model, i.e., $(X, Y) \sim p_{\theta_{N-1}}^{\otimes 2}$. Under the realizability assumption and without mode collapse, this self-refinement paradigm should converge towards the true model $p_{\theta^\star}$. Our setting characterizes the limiting behavior of this iterative process, i.e., preference based on $\ell_{\theta^\star}$ for observations from $p_{\theta^\star}$. Nonetheless, we do not claim the direct applicability of DP MLE for realistic LLM training.

### 4.1. Upper Bound on the Estimation Error

We establish a high-probability upper bound on the estimation error $\max_{\theta \in \mathcal{C}_n} \|\theta - \theta^\star\|$ in the general case. This requires grasping the geometry of $\mathcal{C}_n$ relative to $\theta^\star$.

**Linearized feasibility set.** Since $\mathcal{C}_n$ is defined by nonlinear preference constraint, analyzing its geometry is challenging, and we thus consider a linearized approximation of it. We define the linearized constraint set as

$$\widetilde{\mathcal{C}}_n := \{\theta \in \Theta \mid \forall i \in [n], \ (X_i, Y_i) \notin \widetilde{\mathcal{D}}(\theta^\star, \theta)\},$$

where $\widetilde{\mathcal{D}}(\theta^\star, \theta) := \{(x, y) \in \mathcal{X}^2 \mid \ell_{\theta^\star}(x, y)^2 + \ell_{\theta^\star}(x, y)\langle \theta - \theta^\star, \nabla \ell_{\theta^\star}(x, y)\rangle < 0\}$. This set replaces $\ell_\theta$ with its first-order Taylor expansion around $\theta^\star$, neglecting higher-order terms. A key assumption is that the true constraints are at least as strong as the linearized ones. This ensures $\mathcal{C}_n \subseteq \widetilde{\mathcal{C}}_n$, allowing us to control $\mathcal{C}_n$ via $\widetilde{\mathcal{C}}_n$.

**Assumption 4.4** (Linearization validity). For all $\theta \neq \theta^\star$, $\widetilde{\mathcal{D}}(\theta^\star, \theta) \subseteq \mathcal{D}(\theta^\star, \theta)$.

**Directional analysis and informative constraints.** To quantify the geometry of $\widetilde{\mathcal{C}}_n$ relative to $\theta^\star$, we analyze deviations along directions $u \in \mathcal{S}_{k-1}$. Define the set of informative samples along direction $u$:

$$\mathcal{G}_1(\theta^\star, u) := \{(x, y) \mid \ell_{\theta^\star}(x, y)\langle u, \nabla_{\theta^\star}\ell_{\theta^\star}(x, y)\rangle < 0\}.$$

This set contains observations whose preferences give information along the direction $u$. Assuming that preferences are informative along all directions, we prevent degenerate cases where some directions lack preference information.

**Assumption 4.5** (Informative Preferences). For all $u \in \mathcal{S}_{k-1}$, $\mathbb{P}_{p_{\theta^\star}^{\otimes 2}}(\mathcal{G}_1(\theta^\star, u)) > 0$.

**Deviation bound via minimum informative sample.** Define $R_{n,u}$ as the maximal deviation from $\theta^\star$ within $\widetilde{\mathcal{C}}_n$ along the direction $u$, i.e.,

$$R_{n,u} := \max\{\varepsilon \geq 0 \mid \theta^\star + \varepsilon u \in \widetilde{\mathcal{C}}_n\}.$$

We define the scaling factor

$$\forall (x, y) \in \mathcal{G}_1(\theta^\star, u), \ V_{\theta^\star, u}(x, y) := \frac{\ell_{\theta^\star}(x, y)}{-\langle u, \nabla_{\theta^\star}\ell_{\theta^\star}(x, y)\rangle}.$$

The value $V_{\theta^\star, u}(X_i, Y_i)$ quantifies the amount of information in the preference between $X_i$ and $Y_i$ to discriminate $\theta^\star$ from other parameters on the half-line directed by $u$. The lower $V_{\theta^\star, u}(X_i, Y_i)$ is, the more discriminative is the preference between $X_i$ and $Y_i$. Since $(x, y) \in \mathcal{G}_1(\theta^\star, u) \setminus \widetilde{\mathcal{D}}(\theta^\star, \theta^\star + \varepsilon u)$ if and only if $V_{\theta^\star, u}(x, y) \geq \varepsilon$, we obtain

$$R_{n,u} \leq \min_{i \in [n]}\{V_{\theta^\star, u}(X_i, Y_i) \mid (X_i, Y_i) \in \mathcal{G}_1(\theta^\star, u)\}.$$

Therefore, the maximal deviation $R_{n,u}$ is upper bounded by the minimum of positive random variables. It remains to upper bound the resulting value of this minimum with high probability and conclude provided some regularities hold, e.g., positive density at zero. By analyzing the distribution of $V_{\theta^\star, u}$, we derive the following probabilistic bound.

**Lemma 4.6.** *Suppose Assumption 4.5 hold. For all $u \in \mathcal{S}_{k-1}$, with probability $1 - \delta$,*

$$R_{n,u} \leq F_{\theta^\star, u}^{-1}(\min\{1, \log(1/\delta)/n\}),$$

*with $F_{\theta^\star, u}(\varepsilon) := \mathbb{P}_{p_{\theta^\star}^{\otimes 2}}(V_{\theta^\star, u} \in (0, \varepsilon])$ c.d.f. of $V_{\theta^\star, u}$.*

Since $\max_{\theta \in \mathcal{C}_n} \|\theta - \theta^\star\| \leq \max_{u \in \mathcal{S}_{k-1}} R_{n,u}$, Lemma 4.6 shows that the estimation error can be controlled by the behavior of $F_{\theta^\star, u}^{-1}$ around zero, where $F_{\theta^\star, u}^{-1}(0) = 0$.

**Regularity assumption.** To control $F_{\theta^\star, u}^{-1}$ near zero, the density $F_{\theta^\star, u}'$ should be positive near zero, and we assume control on $(F_{\theta^\star, u}^{-1})''$.

**Assumption 4.7** (Positive density at zero and regularity of inverse c.d.f.)**.** For all $u \in \mathcal{S}_{k-1}$, $F'_{\theta^\star, u}(0) \in (0, +\infty)$ and there exists $(x_{\theta^\star, u}, M_{\theta^\star, u}) \in (0, 1) \times \mathbb{R}_+$ such that $\sup_{x \in [0, x_{\theta^\star, u}]} |(F^{-1}_{\theta^\star, u})''(x)| \leq M_{\theta^\star, u}$.

Using this assumption and $(F^{-1}_{\theta^\star, u})'(0) = 1/F'_{\theta^\star, u}(0)$, the first-order Taylor expansion with remainder yields

$$\forall x \in [0, x_{\theta^\star, u}], \ |F^{-1}_{\theta^\star, u}(x) - x/F'_{\theta^\star, u}(0)| \leq M_{\theta^\star, u} x^2 / 2 \,.$$

This argument leads to our main theorem, directly for $k = 1$ and using a covering argument for $k > 1$.

**Theorem 4.8.** *Suppose Assumptions 4.2, 4.4, 4.5 and 4.7 hold. Let $\delta \in (0, 1)$. Let $\gamma > 0$ and $N(\gamma)$ be the $\gamma$-covering number of $\Theta$ for the norm $\| \cdot \|$. Let $A^{-1}_{\theta^\star} = \min_{u \in \mathcal{S}_{k-1}} F'_{\theta^\star, u}(0)$, $B^{-1}_{\theta^\star} = \min_{u \in \mathcal{S}_{k-1}} x_{\theta^\star, u}$ and $C_{\theta^\star} = \max_{u \in \mathcal{S}_{k-1}} M_{\theta^\star, u}/2$. When $k = 1$, for all $n \geq B_{\theta^\star} \log(2/\delta)$,*

$$\max_{\theta \in \mathcal{C}_n} \|\theta - \theta^\star\| \leq \frac{A_{\theta^\star}}{n} \log(2/\delta) + \frac{C_{\theta^\star}}{n^2} \log(2/\delta)^2 \,,$$

*with probability $1 - \delta$. When $k > 1$, for all $n \geq B_{\theta^\star} \log(N(\gamma)/\delta)$, with probability $1 - \delta$,*

$$\max_{\theta \in \mathcal{C}_n} \|\theta - \theta^\star\| \leq \gamma + \frac{A_{\theta^\star}}{n} \log \frac{N(\gamma)}{\delta} + \frac{C_{\theta^\star}}{n^2} \log \left( \frac{N(\gamma)}{\delta} \right)^2 \,.$$

When $k > 1$, the choice of the optimal parameter $\gamma$ depends on the norm $\| \cdot \|$. Since $\Theta$ is bounded by $B_\Theta$, $N(\gamma)$ is upper bounded by the covering of the ball having diameter $B_\Theta$. As an example, let $N_2(\gamma)$ be the $\varepsilon$-covering number of the unit ball in $\mathbb{R}^k$ for the Euclidean norm. Then, it is known that $\log N_2(\varepsilon) \approx \log(\varepsilon^2 k)/\varepsilon^2$ if $\varepsilon \gtrsim 1/\sqrt{k}$ and $\log N_2(\varepsilon) \approx k \log \frac{1}{\varepsilon^2 k}$ if $\varepsilon \lesssim 1/\sqrt{k}$.[3] Therefore, optimizing over $\gamma$ yields an upper bound on $\max_{\theta \in \mathcal{C}_n} \|\theta - \theta^\star\|_2$ scaling as $\widetilde{\mathcal{O}}(A_{\theta^\star} k/n)$ when $n \gtrsim A_{\theta^\star} k^{3/2}$, and $\widetilde{\mathcal{O}}((A_{\theta^\star}/n)^{1/3})$ otherwise, where $\widetilde{\mathcal{O}}(\cdot)$ hides logarithmic terms. For large sample size compared to the dimension, i.e., $n \gtrsim A_{\theta^\star} k^{3/2}$, we recover a rate of $\widetilde{\mathcal{O}}(1/n)$.

In conclusion, we have derived generic assumptions under which the rate of decay of the estimation error of $\widehat{\theta}_n^{\text{AE}}$ and $\widehat{\theta}_n^{\text{DP}}$ is in $\widetilde{\mathcal{O}}(1/n)$. This is a significant improvement compared to the asymptotic normality of the $\text{SP}_{\text{det}}$ estimator that implies a rate of $\mathcal{O}(1/\sqrt{n})$.

**Positive examples.** While Assumptions 4.4, 4.5 and 4.7 are restrictive, they hold for $\mathcal{F}_{\mathcal{N}, \Sigma}$ (Appendix F) and $\mathcal{F}_{\text{Lap}, b}$ (Appendix G) when using reward $r_\theta = \log p_\theta$. This yields Theorem 4.3. We have $A_{\theta^\star} = 2b$, $B_{\theta^\star} = 8$ and $C_{\theta^\star} = 16b$ for $\mathcal{F}_{\text{Lap}, b}$, and $A_{\theta^\star} = \frac{\pi(d-1)\Gamma(d/2)}{2\Gamma((d-1)/2)} =_{+\infty} \mathcal{O}(\sqrt{d})$ for $\mathcal{F}_{\mathcal{N}, \Sigma}$, hence a rate in $\mathcal{O}(d^{3/2}/n)$ when $n \gg d^2$.

---

[3]E.g., using Gilbert-Varshamov for the lower bound (Gilbert, 1952) and Maurey's empirical method for the upper bound.

**Extended discussions.** In Appendix B, we discuss how to verify or weaken our assumptions (Appendix B.1), the sources of misspecification (Appendix B.2) and other reward models than log-likelihood (Appendix B.3).

# 5. Lower Bound for Deterministic Feedback

In this section, we show that the rate $\mathcal{O}(1/n)$ is minimax optimal (up to a logarithmic factor) by deriving a matching lower bound. The standard approach to minimax lower bounds in estimation relies on Fano-type inequalities and hypothesis testing reductions. However, due to Assumption 4.2, the Kullback-Leibler divergence and $\chi^2$ distance between $q_{\theta^\star}$ and $q_\theta$ are infinite for $\theta \neq \theta^\star$, making these tools ineffective. Instead, we use Assouad's Lemma (Tsybakov, 2009), which provides lower bounds via the total variation distance (defined as $\text{TV}(\mathbb{P}, \mathbb{Q}) := \|\mathbb{P} - \mathbb{Q}\|_1$ for distributions $\mathbb{P}$ and $\mathbb{Q}$). Since TV is not well-behaved for product distributions, we use the squared Hellinger distance, defined as $\text{H}^2(\mathbb{P}, \mathbb{Q}) := \frac{1}{2}\|\sqrt{\mathbb{P}} - \sqrt{\mathbb{Q}}\|_2^2$, which satisfies

$$\text{TV}(\mathbb{P}^{\otimes n}, \mathbb{Q}^{\otimes n}) \leq \sqrt{2\text{H}^2(\mathbb{P}^{\otimes n}, \mathbb{Q}^{\otimes n})} \leq \sqrt{2n\text{H}^2(\mathbb{P}, \mathbb{Q})}.$$

For further analytical convenience, we also employ the Bhattacharyya coefficient, $\text{BC}(\mathbb{P}, \mathbb{Q}) := \|\sqrt{\mathbb{PQ}}\|_1$, which is related to the Hellinger distance by $\text{H}^2(\mathbb{P}, \mathbb{Q}) = 1 - \text{BC}(\mathbb{P}, \mathbb{Q})$. Since $q_{\theta^\star} q_\theta$ is zero for disagreeing preferences, we define the restricted BC as

$$\widetilde{\text{BC}}(\tilde{\theta}, \theta) := \mathbb{E}_{p_{\tilde{\theta}}^{\otimes 2}} \left[ \mathbb{1}\left( \mathcal{D}(\tilde{\theta}, \theta) \cup \mathcal{D}_0(\tilde{\theta}, \theta) \right) \sqrt{p_\theta^{\otimes 2}/p_{\tilde{\theta}}^{\otimes 2}} \right] \,,$$

where $\mathcal{D}_0(\tilde{\theta}, \theta) := \mathcal{G}_0(\tilde{\theta})^{\complement} \triangle \mathcal{G}_0(\theta)^{\complement}$ is the set where the preferences are zero for exactly one parameter.

Lemma 5.1 decomposes the Hellinger distance between two distributions over the preference triplets into the Hellinger distance between sample-only distributions and the disagreement restricted Bhattacharyya coefficient.

**Lemma 5.1.** $\text{H}^2(q_{\tilde{\theta}}, q_\theta) = \widetilde{\text{BC}}(\tilde{\theta}, \theta) + \text{H}^2(p_{\tilde{\theta}}^{\otimes 2}, p_\theta^{\otimes 2})$ *for all $\theta, \tilde{\theta} \in \Theta$.*

As $\text{H}^2(p_{\tilde{\theta}}^{\otimes 2}, p_\theta^{\otimes 2}) \leq 2\text{H}^2(p_{\tilde{\theta}}, p_\theta)$, deriving a lower bound requires controlling $\text{H}^2(p_{\tilde{\theta}}, p_\theta)$ and $\widetilde{\text{BC}}(\tilde{\theta}, \theta)$, hence, we impose the following assumption.

**Assumption 5.2.** There exists positive constants $c_1, c_2$ independent of $k$ and a dimension and problem-dependent scaling function $\alpha_{\mathcal{F}}(k)$ such that for all $\theta, \tilde{\theta} \in \Theta$,

$$\widetilde{\text{BC}}(\tilde{\theta}, \theta) + \text{H}^2(p_{\tilde{\theta}}^{\otimes 2}, p_\theta^{\otimes 2}) \leq \frac{c_1}{\alpha_{\mathcal{F}}(k)} \|\theta - \tilde{\theta}\| + c_2 \|\theta - \tilde{\theta}\|^2 \,.$$

Theorem 5.3 bounds the minimax estimation error.

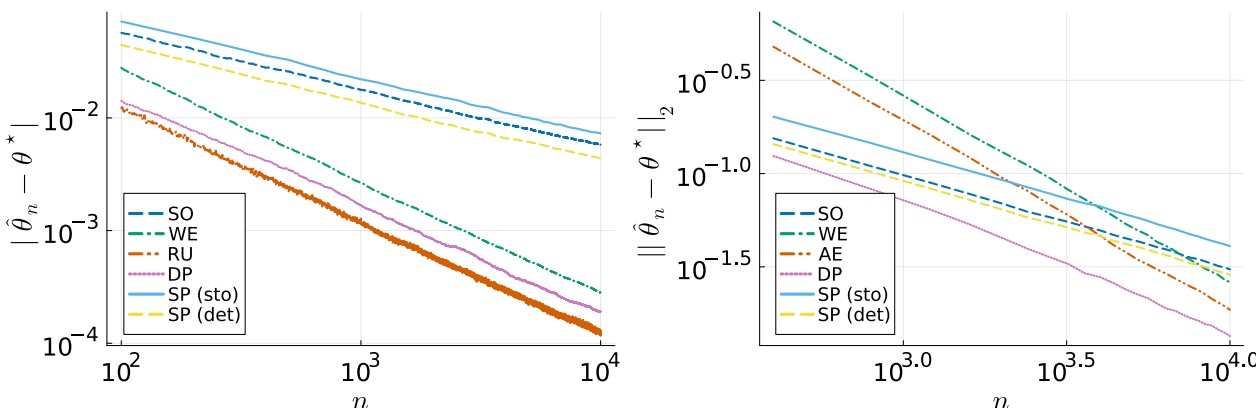

*Figure 1.* Estimation errors for $\mathcal{N}(\theta^\star, I_d)$ where $\theta^\star \sim \mathcal{U}([1,2]^d)$ for (a) $d = 1$ with $N_{\text{runs}} = 10^3$, and (b) $d = 20$ with $N_{\text{runs}} = 10^2$.

**Theorem 5.3.** *Let* $R_{\max} := \inf_{\widehat{\theta}} \sup_{\theta^\star \in \Theta} \mathbb{E}_{q_{\theta^\star}} [\|\widehat{\theta} - \theta^\star\|]$. *Suppose Assumption 5.2 holds. Then,*

$$R_{\max} \geq \Omega \left( \min \left\{ \frac{\alpha_{\mathcal{F}}(k)\sqrt{k}}{n}, \sqrt{\frac{k}{n}} \right\} \right).$$

This result confirms that the $O(1/n)$ rate is minimax optimal up to logarithmic factors. The scaling $\alpha_{\mathcal{F}}(k)$ comes from $\widetilde{\mathrm{BC}}(\theta^\star, \theta)$ (Assumption 5.2), yet it is challenging to link $\alpha_{\mathcal{F}}(k)$ with $A_{\theta^\star}$ without further assumptions.

**Positive examples.** While Assumption 5.2 is restrictive, even when using rewards $r_\theta = \log p_\theta$, it holds for $\mathcal{F}_{\text{Lap},b}$ (Appendix G), i.e.,

$$\widetilde{\mathrm{BC}}(\theta^\star, \theta) = |\theta^\star - \theta|/(2b) + \mathcal{O}(|\theta^\star - \theta|^2),$$

as well as for $\mathcal{F}_{\mathcal{N}, \Sigma}$ (Appendix F), i.e.,

$$\widetilde{\mathrm{BC}}(\theta^\star, \theta) = 2e^{-\|\theta^\star - \theta\|_\Sigma^2/4} F_{\theta^\star, u}(\|\theta^\star - \theta\|_\Sigma),$$

with $F_{\theta^\star, u}(\varepsilon) \leq \varepsilon/A_{\theta^\star}$ and $\alpha_{\mathcal{F}}(d) = A_{\theta^\star}$.

**Dimensionality gap.** While the lower bound in Theorem 5.3 scales as $\Omega(\alpha_{\mathcal{F}}(k)\sqrt{k}/n)$ for $n \geq \alpha_{\mathcal{F}}(k)^2$, the upper bound in Theorem 4.8 scales as $\mathcal{O}(A_{\theta^\star}k/n)$ for $n \gg A_{\theta^\star} k^{3/2}$. Even for the simple case of Gaussian distributions where $A_{\theta^\star} = \alpha_{\mathcal{F}}(d)$, there is a dimensionality gap. Closing this gap is an important direction for future work. Improvements might come from a tighter analysis, e.g., both for the upper and lower bounds, or the derivation of better estimators based on deterministic preferences.

## 6. Experiments

In this section, we compare the empirical performance of the different estimators introduced in this paper. For preferences based on $r_\theta = \log p_\theta$, we conduct a set of experiments

for Gaussian distributions, and defer to Appendix H.1 for experiments on Laplace and Rayleigh distributions. In particular, we consider a uniformly drawn mean parameter $\theta^\star \sim \mathcal{U}([1,2]^d)$ and the isotropic covariance $\Sigma = I_d$. For sample size $n \in [N_{\max}]$ with $N_{\max} = 10^4$, we compute the estimation errors $\|\widehat{\theta}_n - \theta^\star\|_2$. We repeat this process for $N_{\text{runs}}$ different instances and for various choices of $d$.

For $\mathcal{F}_{\mathcal{N}, I_d}$ (Appendix F), the M-estimators can be implemented as $\widehat{\theta}_n^{\text{SO}} = \frac{1}{2n} \sum_{i \in [n]} (X_i + Y_i)$,

$$\widehat{\theta}_n^{\text{SP}} = \arg \min_\theta \|\theta - \widehat{\theta}_n^{\text{SO}}\|_2^2 - \frac{1}{n} \sum_{i \in [n]} \log \sigma(Z_i \ell_\theta(X_i, Y_i)),$$

where $\ell_\theta(X_i, Y_i) = \langle X_i - Y_i, \theta - (X_i + Y_i)/2 \rangle$. Then, the estimators based on $\mathcal{C}_n = \{\theta \mid \forall i \in [n], Z_i \ell_\theta(X_i, Y_i) \geq 0\}$ are $\widehat{\theta}_n^{\text{DP}} = \arg \min_{\theta \in \mathcal{C}_n} \|\theta - \widehat{\theta}_n^{\text{SO}}\|_2^2$ and an arbitrary estimator $\widehat{\theta}_n^{\text{AE}} \in \mathcal{C}_n$. As $\mathcal{C}_n$ is an interval for $d = 1$, we use the randomized uniform (RU) estimator, i.e., $\widehat{\theta}_n^{\text{RU}} \sim \mathcal{U}(\mathcal{C}_n)$. We also consider the worst-case estimator (WE), defined as $\widehat{\theta}_n^{\text{WE}} := \arg \max_{\theta \in \mathcal{C}_n} \|\theta - \theta^\star\|_1$. While it is not a valid estimator due to its $\theta^\star$ dependency, it serves as a proxy for the worst estimation error in $\mathcal{C}_n$.

**Dependency on sample size.** Figure 1(a) confirms empirically the difference in estimation rate between the M-estimators (SO MLE and SP MLE)—obtaining $\mathcal{O}(1/\sqrt{n})$—and our estimators based on $\mathcal{C}_n$—achieving $\mathcal{O}(1/n)$.

However, Figure 1(b) also reveals that the performance of AE and WE deteriorates quickly at small sample sizes when the dimension increases. In contrast, DP MLE consistently outperforms all the other estimators, including SO MLE as theoretically shown in Lemma 4.1.

While $\text{SP}_{\text{det}}$ outperforms SO MLE, Figure 1 also reveals that SP performs worse than SO MLE for finite sample size. Therefore, only an M-estimator based on deterministic preferences improves on sample-only M-estimators empirically.

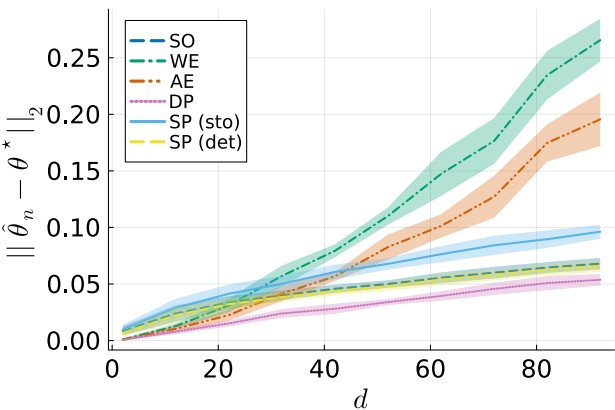

*Figure 2.* Estimation errors as a function of $d$ with $\mathcal{N}(\theta^\star, I_d)$ where $\theta^\star \sim \mathcal{U}([1,2]^d)$, for $n = 10^4$ and $N_{\text{runs}} = 10^3$.

This further highlights the weakness of asymptotic results compared to non-asymptotic guarantees.

While Figure 1(a) suggests that RU and WE perform on par with DP MLE, Figures 1(b) and 2 highlight that DP MLE outperforms WE and AE for larger dimensions, where the gap increases when $d$ is nonnegligible compared to $n$. We conjecture that RU suffers from the same limitation as AE for larger $d$. As empirical evidence, we study other estimators that disentangle the effect of RU's randomness versus its mean behavior, see Appendix H.2.

**Dependency on dimension.** Figure 2 strengthens the aforementioned empirical observations. For fixed sample size and increasing dimension, DP MLE is the only estimator obtaining the best-of-both world estimation error rate, i.e., $\mathcal{O}(\min\{d^{3/2}/n, \sqrt{d/n}\})$.

**Covariance gap.** We show that the covariance gap between SP MLE and SO MLE is relative mild: $(\Delta^{\text{SP}}_{\text{Lap}(0,1)}, \Delta^{\text{SP}_{\text{det}}}_{\text{Lap}(0,1)}) \approx (0.16, 0.08)$ and $(\Delta^{\text{SP}}_{\mathcal{N}(0,1)}, \Delta^{\text{SP}_{\text{det}}}_{\mathcal{N}(0,1)}, R^{\text{SP}_{\text{det}}}_{\mathcal{N}(0,1)}) \approx (0.17, 0.08, 0.10)$. Moreover, $\Delta^{\text{SP}}_{\mathcal{N}(0_d,I_d)}$, $\Delta^{\text{SP}_{\text{det}}}_{\mathcal{N}(0_d,I_d)}$ and $R^{\text{SP}_{\text{det}}}_{\mathcal{N}(0_d,I_d)}$ are close to $\alpha_d I_d$ where $\alpha_d > 0$ is decreasing in $d$ (see Figure 3 in Appendix F). In addition to having a small empirical gap for a moderate value of $n$, the asymptotic gaps between SO MLE and SP MLE are mild.

**Supplementary experiments.** Following the approach of Tang et al. (2024a), we compare estimators using other convex surrogates of the 0-1 loss (Appendix H.3): they all perform similarly. For the logistic loss, we showcase the "mild" impact of normalization and regularization (Appendix H.4).

## 7. Perspectives

This work investigates the role of preference feedback in parameter estimation for continuous parametric distributions. We establish conditions under which preference-based estimators outperform sample-only methods. For stochastic preferences, the preference-based MLE achieves a lower asymptotic variance than its sample-only counterpart. For deterministic preferences, we demonstrate that preference-based estimators can significantly accelerate parameter estimation, achieving an improved $\mathcal{O}(1/n)$ convergence rate compared to the $\mathcal{O}(1/\sqrt{n})$ rate of M-estimators. Our lower bound analysis further confirms that this acceleration is minimax optimal up to dimension-dependent constants.

While our results provide a solid theoretical foundation, several open questions remain. A finer analysis of beyond-M-estimators and their constraint set geometry would allow to better quantify the properties of DP MLE, and provide insights for designing improved estimators that better leverage deterministic preferences. Additionally, exploring alternative preference functions beyond the log-probability gap could extend the applicability of our results.

Finally, a key challenge for future work is to quantify the benefits of preference-based estimation for discrete distributions. For distributions with small support, preference feedback may only localize the unknown parameter within a subset of the simplex, leading to diminishing information gains as the sample size increases. However, understanding how preference-based estimators perform in finite-sample settings, particularly in high-dimensional problems, remains an interesting open problem. Addressing these questions could provide further insights into the role of preferences in machine learning and statistical estimation.

## Acknowledgements

We thank Jaouad Mourtada for insightful early discussions on the univariate Gaussian case. This work was supported by the Swiss National Science Foundation (grant number 212111) and by an unrestricted gift from Google.

## Impact Statement

This paper presents work whose goal is to advance the field of Machine Learning. There are many potential societal consequences of our work, none of which we feel must be specifically highlighted here.

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

## A. Outline

The appendices are organized as follows:

- In Appendix B, we provide detailed discussions on our assumptions, the sources of misspecification and other reward models.

- In Appendix C, we prove the general results presented in Section 3 such as Lemma 3.1.

- In Appendix D, we focus on Section 4 and detail the proofs of Lemma 4.6 and Theorem 4.8

- In Appendix E, we prove the results presented in Section 5.

- In Appendix F, for $\mathcal{F}_{\mathcal{N},\Sigma}$ and preferences based on $r_\theta = \log p_\theta$, we prove all the assumptions introduced in this paper.

- In Appendix G, for $\mathcal{F}_{\mathrm{Lap},b}$ and preferences based on $r_\theta = \log p_\theta$, we prove all the assumptions introduced in this paper.

- In Appendix H, we provide supplementary experiments to support our theoretical findings.

## B. Extended Discussions

We provide detailed discussions on how to verify or weaken our assumptions (Appendix B.1), the sources of misspecification (Appendix B.2) and other reward models than log-likelihood (Appendix B.3).

### B.1. Verifying or Weakening our Assumptions

Since our assumptions are restrictive, it is natural to wonder how they can be verified or weakened.

**Verifying our assumptions.** Even a closed-form definition of $p_\theta$ and $\ell_\theta$ is given, our assumptions are challenging to verify, hence we suggest using a formal verifier (e.g., Lean (mathlib Community, 2020)) or software (e.g., SageMath (The Sage Developers, 2022)). Empirically, they can be confirmed or rejected by sampling from $p_{\theta^\star}^{\otimes 2}$. Assumption 4.4 is rejected by exhibiting $(X_i, Y_i) \in \widetilde{\mathcal{D}}(\theta^\star, \theta) \setminus \mathcal{D}(\theta^\star, \theta)$. Assumptions 4.2 and 4.5 are confirmed by finding $(X_i, Y_i) \in \mathcal{D}(\theta^\star, \theta)$ and $(X_i, Y_i) \in \mathcal{G}_1(\theta^\star, u)$. Those tests' sampling complexity scales as the inverse event's probability. Using Dvoretzky–Kiefer–Wolfowitz inequality (Dvoretzky et al., 1956), $F_{\theta^\star, u}$ can be estimated to verify that Assumption 4.7 holds.

**Restrictive assumptions.** When studying DP MLE only, we conjecture that the "global" Assumptions 4.2 and 4.4 can be weakened to local versions. Using time-uniform concentration results, we can build a sequence of shrinking confidence regions $(\mathcal{R}_n)_n$ around SO MLE that contains $\theta^\star$ for all time $n$ with high probability. Then, we modify DP MLE to be constrained on $\mathcal{R}_n \cap \mathcal{C}_n$. For $n$ large enough and with high probability, $\mathcal{R}_n \cap \mathcal{C}_n$ will be included in a local neighborhood of $\theta^\star$ under which the "local" Assumptions 4.2 and 4.4 are satisfied. Given that Assumption 4.4 is based on "ignoring" the reminder term in a first-order Taylor expansion, assuming a local version is a significantly weaker requirement.

### B.2. Sources of Misspecification

There are several possible sources of misspecification not taken into account by our current analysis.

**Preference model.** The Bradley-Terry model that uses reward-based preferences has limited expressivity as it doesn't allow for intransitive preferences. Even when individuals exhibit transitive preferences, their averaged preferences can be intransitive due to disagreements, see Munos et al. (2024) or Swamy et al. (2024).

**Parameter space.** When $\theta^\star \notin \Theta$, the deterministic preferences might not provide separability within $\Theta$. The definition of DP MLE should be modified to combine the cross-entropy loss and the classification 0-1 loss, i.e.,

$$\widehat{\theta}_n^{\mathrm{DP}} \in \arg\min_{\theta \in \Theta} \left\{ L_n^{\mathrm{SO}}(\theta) + \lambda \sum_{i \in [n]} \mathbb{1}\left(Z_i \ell_\theta(X_i, Y_i) < 0\right) \right\}, \tag{5}$$

where $\lambda > 0$ is a regularization between those two losses. Equation (5) is reminiscent of single-stage alignment procedures such as ORPO (Hong et al., 2024) and ASFT (Wang et al., 2024), see, e.g., Gorbatovski et al. (2025). Without separability, solving Eq. (5) can be NP-hard. Under sufficient regularity, $\widehat{\theta}_n^{\mathrm{DP}}$ converges to $\theta_0 \in \arg\min_{\theta \in \Theta}\{\mathrm{KL}(\theta^\star, \theta) + \lambda m(\theta)\}$ where

$m(\theta) = \mathbb{P}_{p_{\theta^\star}^{\otimes 2}}(\mathcal{D}(\theta^\star, \theta))$ and $\theta_0 \neq \theta^\star$. As $\theta \mapsto m(\theta)$ can be non-convex, computing $\theta_0$ might be challenging. Deriving a tractable ELBO method for this optimization is an interesting direction to obtain tractable and robust estimators. As $\theta_0$ lies in the boundary of $\Theta$, we should control the maximal deviation with respect to $\theta_0$ for directions that point towards the interior of $\Theta$ to prove an accelerated rate. While parts of our analysis could be used, we believe that finer technical arguments are required.

**Parametric model.** The true distribution $p^\star$ of the observations might not even be a member of our class of distributions $\mathcal{F}$, i.e., $p^\star \notin \mathcal{F}$. These situations occur when $\mathcal{F}$ doesn't contain the true structure, e.g., other parametric or non-parametric class of distributions. Then, $L_n^{\mathrm{SO}}(\theta)$ can be interpreted as a quasi-log-likelihood term. Let us denote by SO quasi-MLE the estimator based on SO MLE for this quasi-log-likelihood. Under sufficient regularity, SO quasi-MLE converges towards $\theta_0 \in \arg\min_{\theta \in \Theta} \mathrm{KL}(p^\star, p_\theta)$ where $p^\star \neq p_{\theta_0} \in \mathcal{F}$. Without the separability from well-specified deterministic preference, we define DP quasi-MLE as in Eq. (5). Under sufficient regularity, DP quasi-MLE converges towards the minimizer of a similar optimization problem combining the KL term and a misspecified equivalent of $m(\theta)$.

### B.3. Reward models

Except for Theorem 4.3, all the derivations in Section 4 hold for general (hence reward-based) preference models provided Assumptions 4.2, 4.4, 4.5 and 4.7 hold. Characterizing the expressivity of parametric rewards satisfying those assumptions is interesting, yet challenging. We provide two positive and one negative examples.

**Positive: monotonic reward.** Suppose that $\tilde{\ell}_\theta(x, y) = f(p_\theta(x)) - f(p_\theta(x))$ where $f$ is increasing on $[0, 1]$. Since $\mathrm{sign}(\tilde{\ell}_\theta) = \mathrm{sign}(\ell_\theta)$, hence the parameters with zero classification loss and our estimators are the same. Therefore, our results hold for this class of rewards when our assumptions hold for the log-likelihood reward. When $f$ is decreasing, the preferences are "reversed", and similar arguments can be made. This example includes (1) normalization by a multiplicative constant (e.g., temperature $\beta$) and (2) the odds-ratio reward-based preference based on $f(x) = \log(x/(1-x))$ used by ORPO in Hong et al. (2024).

**Positive: margin with Gaussian.** Suppose that $\tilde{\ell}_\theta = \ell_\theta + c$ where $c$ is a constant and $\ell_\theta$ is the Gaussian log-likelihood preference. By extending our computations from Appendix F, Assumptions 4.2, 4.4, 4.5 and 4.7 hold with $c$-dependent positive constants. Margins are used by SimPO from Meng et al. (2024) and IPO from Azar et al. (2024).

**Negative: reference model with Gaussian.** Suppose that $\tilde{\ell}_\theta = \ell_\theta - \ell_{\theta_0}$ where $\theta_0$ is known and $\ell_\theta$ is the Gaussian log-likelihood preference. Since $\tilde{\ell}_\theta(x, y) = \langle x - y, \theta - \theta_0 \rangle$ and $\nabla_\theta \tilde{\ell}_\theta(x, y) = x - y$, Assumption 4.5 is violated for $u = \theta^\star - \theta_0$, i.e., $\mathbb{P}_{p_{\theta^\star}^{\otimes 2}}(\mathcal{G}_1(\theta^\star, \theta^\star - \theta_0)) = 0$. Not all direct alignment algorithms rely on a reference model, see, e.g., SimPO and ORPO.

## C. Proofs of Section 3

### C.1. Stochastic Preferences

Under enough regularity, by swapping the integration and the differentiation operators, we can show that

$$\mathbb{E}_{q_{\theta, h_{\mathrm{sto}}}}[\nabla_\theta \log q_{\theta, h_{\mathrm{sto}}}] = \nabla_\theta 1 = 0_k \quad \text{and} \quad \mathbb{E}_{q_{\theta, h_{\mathrm{sto}}}}[-\nabla_\theta^2 \log q_{\theta, h_{\mathrm{sto}}}] = \mathbb{E}_{q_{\theta, h_{\mathrm{sto}}}}[\nabla_\theta \log q_{\theta, h_{\mathrm{sto}}} \nabla_\theta \log q_{\theta, h_{\mathrm{sto}}}^\mathsf{T}].$$

Below, we detail the proof of Lemma 3.1.

*Proof.* Direct computation yields that

$$\nabla_{\theta^\star} \log q_{\theta^\star}(x, y, z) = \nabla_{\theta^\star} \log p_{\theta^\star}^{\otimes 2}(x, y) + z\sigma(-z\ell_{\theta^\star}(x, y))\nabla_{\theta^\star}\ell_{\theta^\star}(x, y),$$
$$\nabla_{\theta^\star}^2 \log q_{\theta^\star}(x, y, z) = \nabla_{\theta^\star}^2 \log p_{\theta^\star}^{\otimes 2}(x, y) - \sigma(\ell_{\theta^\star}(x, y))\sigma(-\ell_{\theta^\star}(x, y))\nabla_{\theta^\star}\ell_{\theta^\star}(x, y)\nabla_{\theta^\star}\ell_{\theta^\star}(x, y)^\mathsf{T}$$
$$+ z\sigma(-z\ell_{\theta^\star}(x, y))\nabla_{\theta^\star}^2\ell_{\theta^\star}(x, y).$$

where we used that $g'(x) = \sigma(-x)$ and $g''(x) = -\sigma'(-x) = -\sigma(x)\sigma(-x)$ with $g(x) = \log \sigma(x)$. By definition of $h_{\mathrm{sto}}$,

$$\mathbb{E}_{Z|(X,Y)}\left[Z\sigma(-Z\ell_{\theta^\star}(X, Y))\nabla_{\theta^\star}^2\ell_{\theta^\star}(X, Y)\right]$$
$$= (\sigma(-\ell_{\theta^\star}(X, Y))\sigma(\ell_{\theta^\star}(X, Y)) - \sigma(\ell_{\theta^\star}(X, Y))\sigma(-\ell_{\theta^\star}(X, Y)))\nabla_{\theta^\star}^2\ell_{\theta^\star}(X, Y) = 0_{d \times d}.$$

Therefore, we have $\mathcal{I}(q_{\theta^\star, h_{\mathrm{sto}}}) = \mathcal{I}(p_{\theta^\star}^{\otimes 2}) + \Delta_{\theta^\star}^{\mathrm{SP}}$ with $\Delta_\theta^{\mathrm{SP}} = \mathbb{E}_{p_{\theta^\star}^{\otimes 2}}[\sigma(\ell_\theta)\sigma(-\ell_\theta)\nabla_\theta \ell_\theta (\nabla_\theta \ell_\theta)^\intercal]$. For all $x \in \mathbb{R}^k$, we have

$$x^\intercal \Delta_\theta^{\mathrm{SP}} x = \|x\|^2 \mathbb{E}_{p_{\theta^\star}^{\otimes 2}}[\sigma(\ell_\theta)\sigma(-\ell_\theta)\langle x/\|x\|, \nabla_\theta \ell_\theta \rangle^2] \geq 0 \,.$$

It is direct to see that this inequality is strict except if $\mathbb{P}_{p_{\theta^\star}^{\otimes 2}}(\langle x/\|x\|, \nabla_\theta \ell_\theta \rangle^2 = 0) = 1$. Therefore, $\Delta_{\theta^\star}^{\mathrm{SP}}$ is a positive definite matrix if $\mathbb{P}_{p_{\theta^\star}^{\otimes 2}}(|\langle u, \nabla_{\theta^\star} \ell_{\theta^\star}\rangle| > 0) > 0$ for all $u \in \mathcal{S}_{k-1}$. Note that this condition is implied by Assumption 4.5. $\qquad\square$

## C.2. Deterministic Preferences

**Consistency of** SP. Let $M(\theta) := \mathbb{E}_{p_{\theta^\star}^{\otimes 2}}[\log q_{\theta, h_{\mathrm{sto}}}(X, Y, \mathrm{sign}(\ell_{\theta^\star}(X, Y)))]$. Under enough regularity, we obtain

$$\mathbb{E}_{p_{\theta^\star}^{\otimes 2}}[\nabla_{\theta^\star} \log p_{\theta^\star}^{\otimes 2}] = 0_k \quad \text{and} \quad \nabla_{\theta^\star} M(\theta^\star) = \mathbb{E}_{p_{\theta^\star}^{\otimes 2}}[\mathrm{sign}(\ell_{\theta^\star}(X, Y))\sigma(-|\ell_{\theta^\star}(X, Y)|)\nabla_{\theta^\star} \ell_{\theta^\star}(X, Y)] \,.$$

Therefore, $\theta^\star$ is the unique maximizer of $M(\theta)$ if $\nabla_{\theta^\star} M(\theta^\star) = 0_k$, i.e., if (3) holds true.

**Asymptotic normality of** SP. Provided (3), under enough regularity, the theory of M-estimator yields that

$$\sqrt{n}(\widehat{\theta}_n^{\mathrm{SP\,det}} - \theta^\star) \rightsquigarrow_{n \to +\infty} \mathcal{N}(0_k, V_{1,\theta^\star}^{-1} V_{2,\theta^\star} V_{1,\theta^\star}^{-1}) \,,$$

where

$$V_{1,\theta^\star} = \mathbb{E}_{(X,Y)\sim p_{\theta^\star}^{\otimes 2}}\left[-\nabla_{\theta^\star}^2 \log q_{\theta^\star, h_{\mathrm{sto}}}(X, Y, \mathrm{sign}(\ell_{\theta^\star}(X, Y)))\right] \,,$$
$$V_{2,\theta^\star} = \mathbb{E}_{(X,Y)\sim p_{\theta^\star}^{\otimes 2}}\left[\nabla_{\theta^\star} \log q_{\theta^\star, h_{\mathrm{sto}}}(X, Y, \mathrm{sign}(\ell_{\theta^\star}(X, Y)))\nabla_{\theta^\star} \log q_{\theta^\star, h_{\mathrm{sto}}}(X, Y, \mathrm{sign}(\ell_{\theta^\star}(X, Y)))^\intercal\right] \,.$$

Below we detail the proof of Lemma 3.2.

*Proof.* Combining $z = \mathrm{sign}(\ell_{\theta^\star}(x, y))$ with the same manipulation as above yields

$$\nabla_{\theta^\star} \log q_{\theta^\star, h_{\mathrm{sto}}}(x, y, z) = \nabla_{\theta^\star} \log p_{\theta^\star}^{\otimes 2}(x, y) + \mathrm{sign}(\ell_{\theta^\star}(x, y))\sigma(-|\ell_{\theta^\star}(x, y)|)\nabla_{\theta^\star} \ell_{\theta^\star}(x, y) \,,$$
$$\begin{aligned} \nabla_{\theta^\star} \log q_{\theta^\star, h_{\mathrm{sto}}}(x, y, z)\nabla_{\theta^\star} \log q_{\theta^\star, h_{\mathrm{sto}}}(x, y, z)^\intercal &= \nabla_{\theta^\star} \log p_{\theta^\star}^{\otimes 2}(x, y)\nabla_{\theta^\star} \log p_{\theta^\star}^{\otimes 2}(x, y)^\intercal \\ &+ \sigma(-|\ell_{\theta^\star}(x, y)|)^2 \nabla_{\theta^\star} \ell_{\theta^\star}(x, y)\nabla_{\theta^\star} \ell_{\theta^\star}(x, y)^\intercal \\ &+ \mathrm{sign}(\ell_{\theta^\star}(x, y))\sigma(-|\ell_{\theta^\star}(x, y)|)\left(\nabla_{\theta^\star} \log p_{\theta^\star}^{\otimes 2}(x, y)\nabla_{\theta^\star} \ell_{\theta^\star}(x, y)^\intercal + \nabla_{\theta^\star} \ell_{\theta^\star}(x, y)\nabla_{\theta^\star} \log p_{\theta^\star}^{\otimes 2}(x, y)^\intercal\right) \end{aligned}$$
$$\begin{aligned} \nabla_{\theta^\star}^2 \log q_{\theta^\star, h_{\mathrm{sto}}}(x, y, z) &= \nabla_{\theta^\star}^2 \log p_{\theta^\star}^{\otimes 2}(x, y) - \sigma(\ell_{\theta^\star}(x, y))\sigma(-\ell_{\theta^\star}(x, y))\nabla_{\theta^\star} \ell_{\theta^\star}(x, y)\nabla_{\theta^\star} \ell_{\theta^\star}(x, y)^\intercal \\ &+ \mathrm{sign}(\ell_{\theta^\star}(x, y))\sigma(-|\ell_{\theta^\star}(x, y)|)\nabla_{\theta^\star}^2 \ell_{\theta^\star}(x, y) \,. \end{aligned}$$

where we used that $g'(x) = \sigma(-x)$ and $g''(x) = -\sigma'(-x) = -\sigma(x)\sigma(-x)$ with $g(x) = \log \sigma(x)$. Using that $\sigma(-|\ell_{\theta^\star}(x, y)|)^2 = \sigma(-|\ell_{\theta^\star}(x, y)|) - \sigma(-\ell_{\theta^\star}(x, y))\sigma(\ell_{\theta^\star}(x, y))$, we have

$$V_{1,\theta^\star} = \mathcal{I}(p_{\theta^\star}^{\otimes 2}) + \Delta_{\theta^\star}^{\mathrm{SP}} - H_{\theta^\star}^{\mathrm{SP\,det}} \quad \text{and} \quad V_{2,\theta^\star} = \mathcal{I}(p_{\theta^\star}^{\otimes 2}) + M_{2,\theta^\star} - \Delta_{\theta^\star}^{\mathrm{SP}} - R_{\theta^\star}^{\mathrm{SP\,det}}$$

where $\Delta_\theta^{\mathrm{SP}} = \mathbb{E}_{p_{\theta^\star}^{\otimes 2}}[\sigma(\ell_\theta)\sigma(-\ell_\theta)\nabla_\theta \ell_\theta (\nabla_\theta \ell_\theta)^\intercal]$ as in Lemma 3.1, and we define

$$H_{\theta^\star}^{\mathrm{SP\,det}} = \mathbb{E}_{p_{\theta^\star}^{\otimes 2}}\left[\mathrm{sign}(\ell_{\theta^\star})\sigma(-|\ell_{\theta^\star}|)\nabla_{\theta^\star}^2 \ell_{\theta^\star}\right] \quad, \quad M_{2,\theta^\star} = \mathbb{E}_{p_{\theta^\star}^{\otimes 2}}\left[\sigma(-|\ell_{\theta^\star}|)\nabla_{\theta^\star} \ell_{\theta^\star}(\nabla_{\theta^\star} \ell_{\theta^\star})^\intercal\right] \quad \text{and}$$
$$R_{\theta^\star}^{\mathrm{SP\,det}} = -\mathbb{E}_{p_{\theta^\star}^{\otimes 2}}\left[\mathrm{sign}(\ell_{\theta^\star})\sigma(-|\ell_{\theta^\star}|)\left(\nabla_{\theta^\star} \log p_{\theta^\star}^{\otimes 2}(\nabla_{\theta^\star} \ell_{\theta^\star})^\intercal + \nabla_{\theta^\star} \ell_{\theta^\star}(\nabla_{\theta^\star} \log p_{\theta^\star}^{\otimes 2})^\intercal\right)\right] \,.$$

Using that $\mathcal{I}(q_{\theta^\star, h_{\mathrm{sto}}}) = \mathcal{I}(p_{\theta^\star}^{\otimes 2}) + \Delta_{\theta^\star}^{\mathrm{SP}}$ (Lemma 3.1), $\mathrm{SP}_{\mathrm{det}}$ is asymptotically better than SP if and only if $V_{1,\theta^\star}^{-1} V_{2,\theta^\star} V_{1,\theta^\star}^{-1} \prec \mathcal{I}(q_{\theta^\star, h_{\mathrm{sto}}})^{-1}$. This condition can be rewritten as

$$\mathcal{I}(p_{\theta^\star}^{\otimes 2}) + M_{2,\theta^\star} - \Delta_{\theta^\star}^{\mathrm{SP}} - R_{\theta^\star}^{\mathrm{SP\,det}} \prec \left(\mathcal{I}(p_{\theta^\star}^{\otimes 2}) + \Delta_{\theta^\star}^{\mathrm{SP}} - H_{\theta^\star}^{\mathrm{SP\,det}}\right)\left(\mathcal{I}(q_{\theta^\star, h_{\mathrm{sto}}}) - H_{\theta^\star}^{\mathrm{SP\,det}}\right)\mathcal{I}(q_{\theta^\star, h_{\mathrm{sto}}})^{-1} \,. \qquad (6)$$

The condition (6) heavily depends on the geometry of $p_{\theta^\star}^{\otimes 2}$ and $\ell_{\theta^\star}$, hence it is unreasonable to assume in all generality.

In the following, we consider the special case where $H_{\theta^\star}^{\mathrm{SP}_{\det}} = 0_{d \times d}$. This occurs when $\theta \to \ell_\theta$ is linear, e.g., for $\mathcal{F}_{\mathcal{N}, \Sigma}$ and $\mathcal{F}_{\mathrm{Lap}, b}$ and preferences based on $r_\theta = \log p_\theta$. Then, the condition (6) rewrites as

$$R_{\theta^\star}^{\mathrm{SP}_{\det}} + \Delta_{\theta^\star}^{\mathrm{SP}_{\det}} \succ 0_{d \times d} \quad \text{with} \quad \Delta_{\theta^\star}^{\mathrm{SP}_{\det}} := 2\Delta_{\theta^\star}^{\mathrm{SP}} - M_{2,\theta^\star} \; .$$

Using that $\min_{x \in \mathbb{R}} \sigma(|x|) = 1/2$ achieved only at $x = 0$, we have directly that, for all $x \in \mathbb{R}^k$,

$$x^\intercal \Delta_{\theta^\star}^{\mathrm{SP}_{\det}} x = \|x\|^2 \mathbb{E}_{p_{\theta^\star}^{\otimes 2}}[(2\sigma(|\ell_{\theta^\star}|) - 1)\sigma(-|\ell_{\theta^\star}|)\langle x/\|x\|, \nabla_{\theta^\star}\ell_{\theta^\star}\rangle^2] \geq 0 \; ,$$

It is direct to see that this inequality is strict except if $\mathbb{P}_{p_{\theta^\star}^{\otimes 2}}(\ell_{\theta^\star}\langle x/\|x\|, \nabla_{\theta^\star}\ell_{\theta^\star}\rangle = 0) = 1$. Therefore, $\Delta_{\theta^\star}^{\mathrm{SP}_{\det}}$ is a positive definite matrix if $\mathbb{P}_{p_{\theta^\star}^{\otimes 2}}(|\ell_{\theta^\star}\langle u, \nabla_{\theta^\star}\ell_{\theta^\star}\rangle| > 0) > 0$ for all $u \in \mathcal{S}_{k-1}$. Then, a sufficient condition for the condition (6) to hold is that $R_{\theta^\star}^{\mathrm{SP}_{\det}}$ is a p.s.d. matrix, i.e., $R_{\theta^\star}^{\mathrm{SP}_{\det}} \succeq 0_{d \times d}$.

In summary, we have derived sufficient conditions for $\mathrm{SP}_{\det}$ to be asymptotically better than SP, namely $H_{\theta^\star}^{\mathrm{SP}_{\det}} = 0_{d \times d}$, $R_{\theta^\star}^{\mathrm{SP}_{\det}} \succeq 0_{d \times d}$ and $\mathbb{P}_{p_{\theta^\star}^{\otimes 2}}(|\ell_{\theta^\star}\langle u, \nabla_{\theta^\star}\ell_{\theta^\star}\rangle| > 0) > 0$ for all $u \in \mathcal{S}_{k-1}$. Note that this last condition is implied by Assumption 4.5. $\qquad\square$

## D. Proofs of Section 4
### D.1. Proof of Lemma 4.1

*Proof.* For Gaussian distributions, this is a direct consequence of the following facts: $\theta^\star \in \mathcal{C}_n$, $\widehat{\theta}_n^{\mathrm{DP}} \in \arg\min_{\theta \in \mathcal{C}_n} \|\theta - \widehat{\theta}_n^{\mathrm{SO}}\|_\Sigma^2$ and $\mathcal{C}_n$ is convex. $\qquad\square$

### D.2. Proof of Lemma 4.6

*Proof.* Let $u \in \mathcal{S}_{k-1}$. Let $\widetilde{F}_{\theta^\star, u}$ be the c.d.f. of $V_{\theta^\star, u}(X, Y)$ when $(X, Y) \sim (p_{\theta^\star}^{\otimes 2})_{|\mathcal{G}_1(\theta^\star, u)}$, i.e., $p_{\theta^\star}^{\otimes 2}$ truncated to $\mathcal{G}_1(\theta^\star, u)$. Then, $\widetilde{F}_{\theta^\star, u}(\varepsilon) = F_{\theta^\star, u}(\varepsilon)/\alpha_{\theta^\star, u}$. Let $\alpha_{\theta^\star, u} = \mathbb{P}_{p_{\theta^\star}^{\otimes 2}}(\mathcal{G}_1(\theta^\star, u))$ and $N_{\theta^\star, u} = \sum_{i \in [n]} \mathbb{1}((X_i, Y_i) \in \mathcal{G}_1(\theta^\star, u)) \sim \mathrm{Bin}(n, \alpha_{\theta^\star, u})$. Let $\widetilde{R}_{n,u} = \min_{i \in [n], (X_i, Y_i) \in \mathcal{G}_1(\theta^\star, u)} V_{\theta^\star, u}(X_i, Y_i)$. Using the derivation in Section 4.1, we have that $R_{n,u} \leq \widetilde{R}_{n,u}$. Let $\varepsilon > 0$. Conditioned on $N_{\theta^\star, u}$, it is direct to see that

$$\mathbb{P}(\widetilde{R}_{n,u} > \varepsilon \mid N_{\theta^\star, u}) = 1 - (1 - (1 - \widetilde{F}_{\theta^\star, u}(\varepsilon))^{N_{\theta^\star, u}}) = \left(1 - \widetilde{F}_{\theta^\star, u}(\varepsilon)\right)^{N_{\theta^\star, u}} \; .$$

Using that $N_{\theta^\star, u} \sim \mathrm{Bin}(n, \alpha_{\theta^\star, u})$, $\mathbb{E}_{X \sim \mathrm{Bin}(n,p)}[s^X] = (1 - p + ps)^n$ and $1 - x \leq \exp(-x)$, we obtain that

$$\mathbb{P}(R_{n,u} > \varepsilon) \leq \mathbb{P}(\widetilde{R}_{n,u} > \varepsilon) \leq \left(1 - \alpha_{\theta^\star, u}\widetilde{F}_{\theta^\star, u}(\varepsilon)\right)^n \leq \exp\left(-nF_{\theta^\star, u}(\varepsilon)\right) \; .$$

Taking $\varepsilon = F_{\theta^\star, u}^{-1}(\min\{1, \log(1/\delta)/n\})$ concludes the proof. $\qquad\square$

### D.3. Proof of Theorem 4.8

*Proof.* For all $u \in \mathcal{S}_{k-1}$, let $(A_{\theta^\star}, B_{\theta^\star}, C_{\theta^\star})$ defined as in Theorem 4.8. Since $\mathcal{C}_n \subseteq \widetilde{\mathcal{C}}_n$ under Assumption 4.4, we obtain that $\max_{\theta \in \mathcal{C}_n} \|\theta - \theta^\star\| \leq \max_{\theta \in \widetilde{\mathcal{C}}_n} \|\theta - \theta^\star\|$.

**Case $k = 1$.** Since $|\mathcal{S}_0| = 2$, using Lemma 4.6 with a union bound yield that, with probability at least $1 - \delta$,

$$\max_{\theta \in \widetilde{\mathcal{C}}_n} \|\theta - \theta^\star\| \leq \max_{u \in \mathcal{S}_0} R_{n,u} \leq \max_{u \in \mathcal{S}_0} F_{\theta^\star, u}^{-1}(\min\{1, \log(2/\delta)/n\}) \; .$$

Under Assumption 4.7, for $n \geq B_{\theta^\star} \log(2/\delta)$, we can conclude the proof since

$$n \max_{\theta \in \mathcal{C}_n} \|\theta - \theta^\star\| \leq n \max_{\theta \in \widetilde{\mathcal{C}}_n} \|\theta - \theta^\star\| \leq A_{\theta^\star} \log(2/\delta) + C_{\theta^\star} \log(2/\delta)^2/n \; .$$

**Case $k > 1$.** Let $N(\gamma)$ be the $\gamma$-covering number of $\Theta$ for the norm $\|\cdot\|$. Let $\{\theta_j\}_{j \in [N(\gamma)]}$ be such a $\gamma$-covering. For all $j \in [N(\gamma)]$, let $\varepsilon_j = \|\theta_j - \theta^\star\|$ and $u_j = (\theta_j - \theta^\star)/\varepsilon_j$. Using triangular inequality, we obtain

$$\max_{\theta \in \widetilde{\mathcal{C}}_n} \|\theta - \theta^\star\| \leq \gamma + \max_{j \in [N(\gamma)], \theta_j \in \widetilde{\mathcal{C}}_n} \|\theta_j - \theta^\star\| \leq \gamma + \max_{j \in [N(\gamma)]} \mathbb{1}\left(\theta^\star + \varepsilon_j u_j \in \widetilde{\mathcal{C}}_n\right)\varepsilon_j \leq \gamma + \max_{j \in [N(\gamma)]} R_{n, u_j} \; .$$

Using Lemma 4.6 with a union bound yield that, with probability at least $1 - \delta$,

$$n \max_{\theta \in \mathcal{C}_n} \|\theta - \theta^\star\| \leq n\gamma + \max_{j \in [N(\gamma)]} n F_{\theta^\star, u_j}^{-1} \left( \log(N(\gamma)/\delta)/n \right) \leq n\gamma + A_{\theta^\star} \log(N(\gamma)/\delta) + C_{\theta^\star} \log(N(\gamma)/\delta)^2/n \,.$$

where the last inequality relies on Assumption 4.7 for $n \geq B_{\theta^\star} \log(N(\gamma)/\delta)$. $\qquad\qquad\square$

# E. Proofs of Section 5
## E.1. Proof of Lemma 5.1

*Proof.* It is direct to see that

$$\sqrt{q_{\theta^\star}(x,y,z) q_\theta(x,y,z)} = \begin{cases} 0 & \text{if } (x,y) \in \mathcal{D}(\theta^\star, \theta) \cup \left( \mathcal{G}_0(\theta^\star)^\complement \triangle \mathcal{G}_0(\theta)^\complement \right) \\ \sqrt{p_{\theta^\star}(x) p_{\theta^\star}(y) p_\theta(x) p_\theta(y)} & \text{otherwise} \end{cases} \,.$$

Therefore, we have

$$\mathrm{BC}(q_{\theta^\star}, q_\theta) = \int_{(x,y) \notin \mathcal{D}(\theta^\star,\theta) \cup \left( \mathcal{G}_0(\theta^\star)^\complement \triangle \mathcal{G}_0(\theta)^\complement \right)} \sqrt{p_{\theta^\star}(x) p_{\theta^\star}(y) p_\theta(x) p_\theta(y)} \mathrm{d}x \mathrm{d}y = \mathrm{BC}(p_{\theta^\star}^{\otimes 2}, p_\theta^{\otimes 2}) - \widetilde{\mathrm{BC}}(\theta^\star, \theta) \,.$$

Using that $\mathrm{H}^2(\mathbb{P}, \mathbb{Q}) = 1 - \mathrm{BC}(\mathbb{P}, \mathbb{Q})$, we conclude the proof. $\qquad\qquad\square$

## E.2. Proof of Theorem 5.3

Consider the hypercube $\Theta' = \{\theta_b = \delta b : b \in \{0,1\}^d\} \subseteq \Theta$. Note that $\|\theta_b - \theta_{b'}\| \geq \frac{\delta}{\sqrt{k}} d_H(b, b')$, where $d_H(b, b')$ denotes the Hamming distance between $b$ and $b'$. Then using Assouad's lemma we have

$$R_{\max} \geq \frac{\delta\sqrt{k}}{4} \left( 1 - \max_{d_H(b,b')=1} TV(q_{\theta_b}^{\otimes n}, q_{\theta_{b'}}^{\otimes n}) \right) \,.$$

Upper bounding TV with $\mathrm{H}^2$, Lemma 5.1 yields

$$TV(q_{\theta_b}^{\otimes n}, q_{\theta_{b'}}^{\otimes n}) \leq \sqrt{n(\widetilde{\mathrm{BC}}(\theta_b, \theta_{b'}) + \mathrm{H}^2(p_{\theta_b}^{\otimes 2}, p_{\theta_{b'}}^{\otimes 2}))} \,.$$

Then, Assumption 5.2 implies

$$R_{\max} \geq \frac{\delta\sqrt{k}}{4} \left( 1 - \sqrt{n \left( \frac{c_1 \delta}{\alpha_\mathcal{F}(k)} + 2c_2 \delta^2 \right)} \right) \,.$$

Picking $\delta = \frac{1}{2(c_1 + 2c_2)} \min\{\frac{\alpha_\mathcal{F}(k)}{n}, \frac{1}{\sqrt{n}}\}$ ensures that the term in parenthesis is always greater than $1/2$, hence

$$R_{\max} \geq \frac{\sqrt{k}}{8(c_1 + 2c_2)} \min\left\{ \frac{\alpha_\mathcal{F}(k)}{n}, \frac{1}{\sqrt{n}} \right\} \,.$$

# F. Multivariate Gaussian with Known Covariance

In the following, $\theta = \Sigma^{-1} \mu$ denote the natural parameter of multivariate Gaussian with known covariance matrix $\Sigma$. We have $\mathcal{X} = \mathbb{R}^d$ and $k = d$. Let $\theta \in \Theta$ and $u \in \mathcal{S}_{d-1}$ for the norm $\|\cdot\|_\Sigma$, i.e., $\|u\|_\Sigma = 1$. Let $\mathcal{S}_{2,d-1} = \{x \in \mathbb{R}^d \mid \|u\|_2 = 1\}$. It is direct to see that

$$\ell_\theta(x,y) = \log \frac{p_\theta(x)}{p_\theta(y)} = \langle x - y, \theta - \Sigma^{-1}(x+y)/2 \rangle \quad \text{and} \quad \nabla_{\theta^\star} \ell_{\theta^\star}(x,y) = x - y \,.$$

Therefore, we have

$$\mathcal{G}_0(\theta^\star) = \{(x,y) \in (\mathbb{R}^d)^2 \mid |\langle x - y, \theta^\star - (x+y)/2 \rangle| > 0\} \,,$$

$$\mathcal{G}_1(\theta^\star) = \{(x,y) \in \mathcal{G}_0(\theta^\star) \mid \|x - y\| > 0\} \,,$$

$$\mathcal{D}(\theta^\star, \theta) = \{(x,y) \in (\mathbb{R}^d)^2 \mid \langle x - y, \theta^\star - \Sigma^{-1}(x+y)/2 \rangle^2 + \langle x - y, \theta^\star - \Sigma^{-1}(x+y)/2 \rangle \langle \theta - \theta^\star, x - y \rangle < 0\} \,,$$

$$\mathcal{G}_1(\theta^\star, u) = \{(x,y) \in (\mathbb{R}^d)^2 \mid \langle x - y, \theta^\star - \Sigma^{-1}(x+y)/2 \rangle \langle u, x - y \rangle < 0\} \,,$$

$$\forall (x,y) \in \mathcal{G}_1(\theta^\star, u), \quad V_{\theta^\star, u}(x,y) = \frac{\langle x - y, \Sigma^{-1}(x+y)/2 - \theta^\star \rangle}{\langle u, x - y \rangle} \,.$$

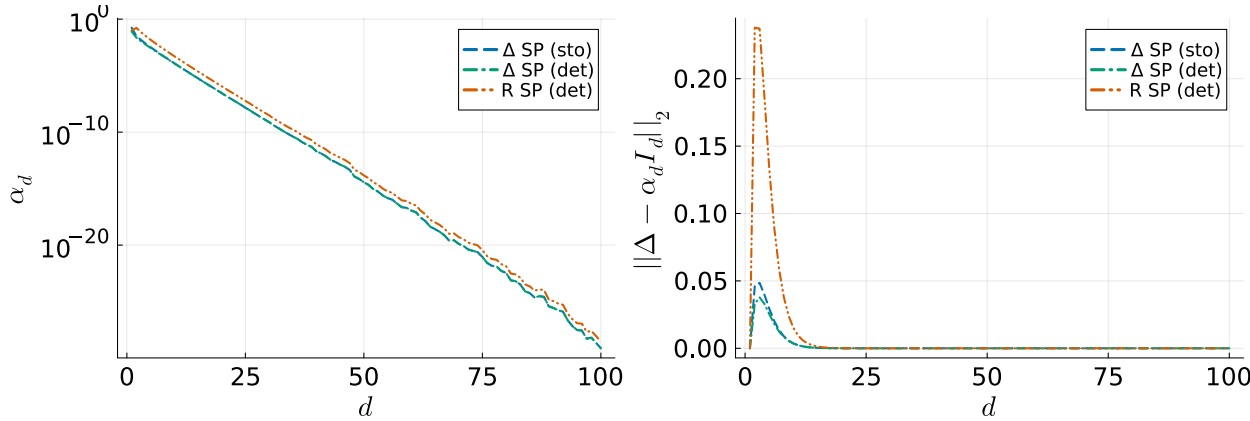

*Figure 3.* Approximations of $\Delta^{\mathrm{SP}}_{\mathcal{N}(0_d,I_d)}$, $\Delta^{\mathrm{SP_{det}}}_{\mathcal{N}(0_d,I_d)}$ and $R^{\mathrm{SP_{det}}}_{\mathcal{N}(0_d,I_d)}$ by (a) $\alpha_d I_d$ and (b) associated error for varying $d$. $N_{\mathrm{runs}} = 10^6$.

**Proof that** $\mathbb{P}_{p_{\theta^\star}^{\otimes 2}}(\mathcal{G}_1(\theta^\star)) > 0$. It is direct to see that $\dim(\mathcal{G}_0(\theta^\star)^\complement) < 2d$ and $\dim(\mathcal{G}_0(\theta^\star) \setminus \mathcal{G}_1(\theta^\star)) < 2d$. Given that $p_{\theta^\star}^{\otimes 2}$ is a continuous distribution on $(\mathbb{R}^d)^2$, we obtain that $\mathbb{P}_{p_{\theta^\star}^{\otimes 2}}(\mathcal{G}_1(\theta^\star)) = \mathbb{P}_{p_{\theta^\star}^{\otimes 2}}(\mathcal{G}_0(\theta^\star)) = 1$.

**Condition in Lemma 3.1.** The condition of Lemma 3.1 is implied by Assumption 4.5, hence we refer to the proof of this result below. Therefore, we have $\mathcal{I}(q_{\theta^\star, h_{\mathrm{sto}}}) \succ \mathcal{I}(p_{\theta^\star}^{\otimes 2})$.

**Consistency of** $\mathrm{SP_{det}}$. To study $\mathrm{SP_{det}}$ for $\mathcal{F}_{\mathcal{N}, \Sigma}$, we use the change of variable $D = \Sigma^{-1/2}(X - Y)/\sqrt{2}$ and $S = \sqrt{2}\Sigma^{-1/2}(\Sigma\theta^\star - (X + Y)/2)$. Then, we have $(D, S) \sim \mathcal{N}(0_{2d}, I_{2d})$ and

$$\ell_{\theta^\star}(X, Y) = \langle S, D \rangle \quad , \quad \nabla_{\theta^\star}\ell_{\theta^\star}(X, Y) = \sqrt{2}\Sigma^{1/2}D \quad , \quad \nabla_{\theta^\star}\log p_{\theta^\star}^{\otimes 2}(X, Y) = 2\Sigma\theta^\star - \sqrt{2}\Sigma^{1/2}S .$$

Let $M(D, S) = \mathrm{sign}(\langle D, S \rangle)\sigma(-|\langle D, S \rangle|)D$. Then, $M(-D, -S) = -M(D, S)$ for all $(D, S) \in \mathbb{R}^{2d}$. By integration of an odd function with respect to $0_d$ with a symmetric distribution around $0_{2d}$, we obtain $\mathbb{E}_{(D,S)\sim\mathcal{N}(0_{2d}, I_{2d})}[M(D, S)] = 0_d$. Therefore, the condition (3) is satisfied and the $\mathrm{SP_{det}}$ is a consistent estimator.

**Asymptotic variance of** $\mathrm{SP_{det}}$. Let $H^{\mathrm{SP_{det}}}_{\theta^\star}$ and $R^{\mathrm{SP_{det}}}_{\theta^\star}$ defined in Lemma 3.2. Since $\ell_\theta(x, y) = \langle x - y, \theta - (x + y)/2 \rangle$ is linear in $\theta$, we have $\nabla^2_{\theta^\star}\ell_{\theta^\star} = 0_{d\times d}$ and $H^{\mathrm{SP_{det}}}_{\theta^\star} = 0_{d\times d}$. The condition $\mathbb{P}_{p_{\theta^\star}^{\otimes 2}}(|\ell_{\theta^\star}\langle u, \nabla_{\theta^\star}\ell_{\theta^\star}\rangle| > 0) > 0$ for all $u \in \mathcal{S}_{d-1}$ is implied by Assumption 4.5, hence we refer to the proof of this result below. Then, the condition $R^{\mathrm{SP_{det}}}_{\theta^\star} \succeq 0_{d\times d}$ is equivalent to $M_3 \succeq 0_{d\times d}$ where

$$M_3 = \mathbb{E}_{(D,S)\sim\mathcal{N}(0_{2d}, I_{2d})}\left[\mathrm{sign}(\langle D, S \rangle)\sigma(-|\langle D, S \rangle|)(DS^\mathsf{T} + SD^\mathsf{T})\right] .$$

When $d = 1$, we have $M_3 = 2\mathbb{E}_{(D,S)\sim\mathcal{N}(0_2, I_2)}[\sigma(-|D, S|)|DS|] > 0$. When $d > 1$, for all $u \in \mathcal{S}_{d-1}$, we have

$$u^\mathsf{T}M_3 u = 2\mathbb{E}_{(D,S)\sim\mathcal{N}(0_{2d}, I_{2d})}\left[\mathrm{sign}(\langle D, S \rangle)\sigma(-|\langle D, S \rangle|)\langle u, D \rangle\langle u, S \rangle\right] ,$$

By rotational symmetry of $\mathcal{N}(0_{2d}, I_{2d})$ and the function to be integrated, showing that $\min_{u\in\mathcal{S}_{d-1}} u^\mathsf{T}M_3 u \geq 0$ is equivalent to showing that $e_1^\mathsf{T}M_3 e_1 \geq 0$, i.e.,

$$\mathbb{E}_{(D,S)\sim\mathcal{N}(0_{2d}, I_{2d})}\left[\mathrm{sign}(\langle D, S \rangle)\sigma(-|\langle D, S \rangle|)D_1 S_1\right] .$$

By symmetry, we conjecture that $\Delta^{\mathrm{SP}}_{\mathcal{N}(0_d,I_d)}$, $\Delta^{\mathrm{SP_{det}}}_{\mathcal{N}(0_d,I_d)}$ and $R^{\mathrm{SP_{det}}}_{\mathcal{N}(0_d,I_d)}$ are of the form $\alpha_d I_d$ where $\alpha_d > 0$ is decreasing in $d$. Figure 3 validates this conjecture numerically.

Using the sufficient condition derived in Appendix C.2, we have shown that $\mathrm{SP_{det}}$ is asymptotically better than SP.

**Proof of Assumption 4.4.** Since $\ell_\theta(x, y) = \langle x - y, \theta - (x + y)/2 \rangle$ is linear in $\theta$, we have $\mathcal{D}(\theta^\star, \theta) = \widetilde{\mathcal{D}}(\theta^\star, \theta)$.

**Proof of Assumption 4.5** For $(X, Y) \sim p_{\theta^\star}^{\otimes 2}$, let $D = \Sigma^{-1/2}(X - Y)/\sqrt{2}$ and $S = \sqrt{2}\Sigma^{-1/2}((X + Y)/2 - \Sigma\theta^\star)$. Then, we have $(D, S) \sim \mathcal{N}(0_{2d}, I_{2d})$. Defining $U = D/\|D\|$, we have $U \sim \mathcal{U}(\mathcal{S}_{2,d-1})$ is independent of $S$. Since $U = D/\|D\| \sim \mathcal{U}(\mathcal{S}_{2,d-1})$ and $\Sigma^{-1}(x + y)/2 - \theta^\star = \Sigma^{-1/2}S/\sqrt{2}$, we obtain

$$\mathbb{P}_{(X,Y)\sim p_{\theta^\star}^{\otimes 2}}((X, Y) \in \mathcal{G}_1(\theta^\star, u)) = \mathbb{P}_{(U,S)\sim\mathcal{U}(\mathcal{S}_{2,d-1})\otimes\mathcal{N}(0_d, I_d)}\left(\langle U, S\rangle\langle u, \Sigma^{1/2}U\rangle > 0\right)$$

$$= \mathbb{P}_{U\sim\mathcal{U}(\mathcal{S}_{2,d-1})}\left(\langle\Sigma^{1/2}u, U\rangle > 0\right)/2 + \mathbb{P}_{U\sim\mathcal{U}(\mathcal{S}_{2,d-1})}\left(\langle\Sigma^{1/2}u, U\rangle < 0\right)/2 = 1/2\,.$$

where we used that, conditioned on $U$, $\langle U, S\rangle \sim \mathcal{N}(0, 1)$ and $\mathbb{P}_{X\sim\mathcal{N}(0,1)}(X < 0) = \mathbb{P}_{X\sim\mathcal{N}(0,1)}(X > 0) = 1/2$. The last equality uses that $\mathbb{P}_{U\sim\mathcal{U}(\mathcal{S}_{2,d-1})}\left(\langle\Sigma^{1/2}u, U\rangle > 0\right) = \mathbb{P}_{U\sim\mathcal{U}(\mathcal{S}_{2,d-1})}\left(\langle\Sigma^{1/2}u, U\rangle < 0\right) = 1/2$ by symmetry of the uniform distribution. Therefore, we have shown that $\mathbb{P}_{p_{\theta^\star}^{\otimes 2}}(\mathcal{G}_1(\theta^\star, u)) = 1/2$ for all $u \in \mathcal{S}_{2,d-1}$.

**Proof of Assumption 4.7.** Let us define $v = \Sigma^{1/2}u$, hence $v \in \mathcal{U}(\mathcal{S}_{2,d-1})$. Let $\Phi$ denote the c.d.f. of $\mathcal{N}(0, 1)$ and $\mathrm{erf}(x) = 2\Phi(x\sqrt{2}) - 1$ be the error function. Let $\varepsilon > 0$. Similarly as above, we obtain that

$$F_{\theta^\star,u}(\varepsilon) = \mathbb{P}_{(X,Y)\sim p_{\theta^\star}^{\otimes 2}}(0 < V_{\theta^\star,u}(X, Y) \le \varepsilon)$$

$$= \mathbb{P}_{(U,S)\sim\mathcal{U}(\mathcal{S}_{2,d-1})\otimes\mathcal{N}(0_d, I_d)}\left(0 < \frac{\langle U, S\rangle}{\langle v, U\rangle} \le \sqrt{2}\varepsilon\right)$$

$$= \frac{1}{2}\mathbb{E}_{U\sim\mathcal{U}(\mathcal{S}_{2,d-1})}\left[2\Phi\left(\sqrt{2}\varepsilon|\langle v, U\rangle|\right) - 1\right] = \frac{1}{2}\mathbb{E}_{U\sim\mathcal{U}(\mathcal{S}_{2,d-1})}\left[\mathrm{erf}\left(\varepsilon|\langle v, U\rangle|\right)\right]\,.$$

where we use conditioning by $U$ as above. By change of variable, we obtain that

$$F_{\theta^\star,u}(\varepsilon) = \frac{1}{\sqrt{\pi}}\mathbb{E}_{U\sim\mathcal{U}(\mathcal{S}_{2,d-1})}\left[\int_0^{\varepsilon|\langle v,U\rangle|} e^{-t^2}\mathrm{d}t\right] = \frac{\varepsilon}{\sqrt{\pi}}\mathbb{E}_{U\sim\mathcal{U}(\mathcal{S}_{2,d-1})}\left[|\langle v, U\rangle|\int_0^1 e^{-x^2\varepsilon^2\langle v,U\rangle^2}\mathrm{d}x\right]\,.$$

Using that $1 - x^2 \le e^{-x^2} \le 1$, we obtain that

$$0 \ge \frac{\sqrt{\pi}}{\varepsilon}F_{\theta^\star,u}(\varepsilon) - \mathbb{E}_{U\sim\mathcal{U}(\mathcal{S}_{2,d-1})}\left[|\langle v, U\rangle|\right] \ge -\frac{\varepsilon^2}{3}\mathbb{E}_{U\sim\mathcal{U}(\mathcal{S}_{2,d-1})}\left[|\langle v, U\rangle|^3\right]\,.$$

Using that

$$\int_0^1 x(-2x\varepsilon^2\langle v, U\rangle^2)e^{-x^2\varepsilon^2\langle v,U\rangle^2}\mathrm{d}x = e^{-\varepsilon^2\langle v,U\rangle^2} - \int_0^1 e^{-x^2\varepsilon^2\langle v,U\rangle^2}\mathrm{d}x\,,$$

we obtain

$$F'_{\theta^\star,u}(\varepsilon) = \frac{1}{\sqrt{\pi}}\mathbb{E}_{U\sim\mathcal{U}(\mathcal{S}_{2,d-1})}\left[|\langle v, U\rangle|e^{-\varepsilon^2\langle v,U\rangle^2}\right] \quad\text{and}\quad F''_{\theta^\star,u}(\varepsilon) = \frac{2\varepsilon}{\sqrt{\pi}}\mathbb{E}_{U\sim\mathcal{U}(\mathcal{S}_{2,d-1})}\left[|\langle v, U\rangle|^3 e^{-\varepsilon^2\langle v,U\rangle^2}\right]\,.$$

Therefore, using Lemma F.1, we have

$$F'_{\theta^\star,u}(0) = \frac{1}{\sqrt{\pi}}\mathbb{E}_{U\sim\mathcal{U}(\mathcal{S}_{2,d-1})}\left[|\langle v, U\rangle|\right] = \frac{2}{d-1}\frac{\Gamma(d/2)}{\pi\Gamma((d-1)/2)} =_{d\to+\infty} \mathcal{O}(1/\sqrt{d})$$

Let us define

$$\varepsilon_{\theta^\star,u} = \sqrt{\frac{\mathbb{E}_{U\sim\mathcal{U}(\mathcal{S}_{2,d-1})}\left[|\langle v, U\rangle|\right]}{2\mathbb{E}_{U\sim\mathcal{U}(\mathcal{S}_{2,d-1})}\left[|\langle v, U\rangle|^3\right]}} \quad\text{and}\quad M_{\theta^\star,u} = 4\pi\sqrt{\frac{\mathbb{E}_{U\sim\mathcal{U}(\mathcal{S}_{2,d-1})}\left[|\langle v, U\rangle|^3\right]}{\mathbb{E}_{U\sim\mathcal{U}(\mathcal{S}_{2,d-1})}\left[|\langle v, U\rangle|\right]^5}}\,.$$

Then, for all $\varepsilon \in (0, \varepsilon_{\theta^\star,u}]$, we obtain that

$$\frac{F''_{\theta^\star,u}(\varepsilon)}{F'_{\theta^\star,u}(\varepsilon)^3} = \frac{\pi\varepsilon}{2}\frac{\mathbb{E}_{U\sim\mathcal{U}(\mathcal{S}_{2,d-1})}\left[|\langle v, U\rangle|^3 e^{-\varepsilon^2\langle v,U\rangle^2}\right]}{\mathbb{E}_{U\sim\mathcal{U}(\mathcal{S}_{2,d-1})}\left[|\langle v, U\rangle|e^{-\varepsilon^2\langle v,U\rangle^2}\right]^3}$$

$$\le \frac{\pi\varepsilon}{2}\frac{\mathbb{E}_{U\sim\mathcal{U}(\mathcal{S}_{2,d-1})}\left[|\langle v, U\rangle|^3\right]}{\left(\mathbb{E}_{U\sim\mathcal{U}(\mathcal{S}_{2,d-1})}\left[|\langle v, U\rangle|\right] - \varepsilon^2\mathbb{E}_{U\sim\mathcal{U}(\mathcal{S}_{2,d-1})}\left[|\langle v, U\rangle|^3\right]\right)^3} \le M_{\theta^\star,u}\,.$$

Since we have $(F_{\theta^\star,u}^{-1})''(x) = -\frac{F_{\theta^\star,u}''(F_{\theta^\star,u}^{-1}(x))}{F_{\theta^\star,u}'(F_{\theta^\star,u}^{-1}(x))^3}$, we obtain

$$\sup_{x\in(0,x_{\theta^\star,u}]} |(F_{\theta^\star,u}^{-1})''(x)| \le M_{\theta^\star,u} \quad \text{where} \quad x_{\theta^\star,u} = F_{\theta^\star,u}(\varepsilon_{\theta^\star,u}) \le \sqrt{\frac{\mathbb{E}_{U\sim\mathcal{U}(\mathcal{S}_{2,d-1})}\left[|\langle\Sigma^{1/2}u,U\rangle|\right]^3}{2\pi\mathbb{E}_{U\sim\mathcal{U}(\mathcal{S}_{2,d-1})}\left[|\langle\Sigma^{1/2}u,U\rangle|^3\right]}} .$$

**Proof of Assumption 4.2.** Let $\varepsilon = \|\theta^\star - \theta\|$ and $u = (\theta^\star - \theta)/\varepsilon$. Then, we have

$$\mathbb{P}_{p_{\theta^\star}^{\otimes 2}}(\mathcal{D}(\theta^\star,\theta)) = \mathbb{P}_{p_{\theta^\star}^{\otimes 2}}(\mathcal{D}(\theta^\star,\theta^\star+\varepsilon u)) \ge \mathbb{P}_{p_{\theta^\star}^{\otimes 2}}(\mathcal{D}(\theta^\star,\theta^\star+\varepsilon u) \cap \mathcal{G}_1(\theta^\star,u)) = \mathbb{P}_{(X,Y)\sim p_{\theta^\star}^{\otimes 2}}(0 < V_{\theta^\star,u}(X,Y) < \varepsilon)$$

Using the above computation, we obtain that $\mathbb{P}_{(X,Y)\sim p_{\theta^\star}^{\otimes 2}}(0 < V_{\theta^\star,u}(X,Y) < \varepsilon) > 0$, hence $\mathbb{P}_{p_{\theta^\star}^{\otimes 2}}(\mathcal{D}(\theta^\star,\theta)) > 0$.

**Proof of Assumption 5.2.** Using that $1 - e^{-x} \le x$, we obtain

$$\mathrm{H}^2(p_{\theta^\star},p_\theta) = 1 - \exp\left(-\frac{1}{8}\|\theta^\star - \theta\|_\Sigma^2\right) \le \frac{1}{8}\|\theta^\star - \theta\|_\Sigma^2 .$$

First, we notice that $\dim\left(\mathcal{G}_0(\theta^\star)^\complement \triangle \mathcal{G}_0(\theta)^\complement\right) < 2d$, hence we can show that

$$\int_{(x,y)\in\mathcal{G}_0(\theta^\star)^\complement\triangle\mathcal{G}_0(\theta)^\complement} \sqrt{p_{\theta^\star}(x)p_{\theta^\star}(y)p_\theta(x)p_\theta(y)}\mathrm{d}x\mathrm{d}y = 0 .$$

Second, we see that

$$\|x - \Sigma\theta^\star\|_{\Sigma^{-1}}^2 + \|y - \Sigma\theta^\star\|_{\Sigma^{-1}}^2 + \|x - \Sigma\theta\|_{\Sigma^{-1}}^2 + \|y - \Sigma\theta\|_{\Sigma^{-1}}^2$$
$$= \|x - y\|_{\Sigma^{-1}}^2 + \|\theta - \theta^\star\|_\Sigma^2 + \|x + y - \Sigma(\theta^\star + \theta)\|_{\Sigma^{-1}}^2 ,$$
$$\mathcal{D}(\theta^\star,\theta) = \left\{(x,y)\in(\mathbb{R}^d)^2 \mid \langle x - y, \theta^\star - \Sigma^{-1}(x+y)/2\rangle^2 + \langle x - y, \theta^\star - \Sigma^{-1}(x+y)/2\rangle\langle\theta - \theta^\star, x - y\rangle < 0\right\} ,$$

Then, we consider the change of variable $u = \Sigma^{-1/2}(x - y)$ and $v = \Sigma^{-1/2}(x + y)$, whose Jacobian has $\det(\Sigma)2^{-d}$ as absolute value of its determinant. Therefore, we obtain

$$e^{\frac{1}{4}\|\theta-\theta^\star\|_\Sigma^2}\widetilde{\mathrm{BC}}(\theta^\star,\theta) = \frac{1}{(4\pi)^d}\int_{(u,v)} \mathbb{1}\left(0 < -\frac{\langle u,\Sigma^{1/2}\theta^\star - v/2\rangle}{\langle\Sigma^{1/2}(\theta-\theta^\star),u\rangle} < 1\right) e^{-\frac{1}{4}\|u\|^2 - \frac{1}{4}\|v-\Sigma^{1/2}(\theta+\theta^\star)\|^2}\mathrm{d}u\mathrm{d}v$$
$$= \frac{1}{(2\pi)^d}\int_{(\tilde{u},\tilde{v})} \mathbb{1}\left(\left|\frac{\langle\tilde{u},\tilde{v}\rangle}{\langle\Sigma^{1/2}(\theta-\theta^\star),\tilde{u}\rangle}\right| < \sqrt{2}\right) e^{-\frac{1}{2}\|\tilde{u}\|^2 - \frac{1}{2}\|\tilde{v}\|^2}\mathrm{d}\tilde{u}\mathrm{d}\tilde{v}$$
$$= \mathbb{P}_{(X,Y)\sim\mathcal{N}(0_d,I_d)^{\otimes 2}}\left(\left|\frac{\langle X,Y\rangle}{\langle\Sigma^{1/2}(\theta-\theta^\star),X\rangle}\right| < \sqrt{2}\right)$$
$$= \mathbb{E}_{U\sim\mathcal{U}(\mathcal{S}_{2,d-1})}\left[\mathrm{erf}\left(|\langle\Sigma^{1/2}(\theta-\theta^\star),U\rangle|\right)\right] = 2F_{\theta^\star,u}(\varepsilon)$$

where the second equality uses the change of variable $\tilde{u} = u/\sqrt{2}$ and $\tilde{v} = (v - \Sigma^{1/2}(\theta+\theta^\star))/\sqrt{2}$, whose Jacobian has determinant $2^d$. The third and the fourth re-uses computation done previously with $\varepsilon = \|\theta - \theta^\star\|_\Sigma$ and $u = (\theta - \theta^\star)/\varepsilon$. Using Lemma F.1 and the above upper bound on $F_{\theta^\star,u}(\varepsilon)$, we obtain

$$\widetilde{\mathrm{BC}}(\theta^\star,\theta) = 2e^{-\varepsilon^2/4}F_{\theta^\star,u}(\varepsilon) \le \frac{4}{d-1}\frac{\Gamma(d/2)}{\pi\Gamma((d-1)/2)}\|\theta^\star - \theta\|_\Sigma .$$

**Lemma F.1.** *Let $\Gamma$ be the $\Gamma$ function. Then,*

$$\forall u \in \mathcal{S}_{2,d-1}, \quad \mathbb{E}_{U\sim\mathcal{U}(\mathcal{S}_{2,d-1})}\left[|\langle u,U\rangle|\right] = \frac{2}{d-1}\frac{\Gamma(d/2)}{\sqrt{\pi}\Gamma((d-1)/2)} =_{d\to+\infty} \mathcal{O}(1/\sqrt{d}) .$$

*Proof.* Due to rotational symmetry of the distribution, for any unit vector $u$,

$$\mathbb{E}_{U\sim\mathcal{U}(\mathcal{S}_{2,d-1})}\left[|\langle u,U\rangle|\right] = \mathbb{E}_{U\sim\mathcal{U}(\mathcal{S}_{2,d-1})}\left[|\langle e_1,U\rangle|\right] = \mathbb{E}_{U\sim\mathcal{U}(\mathcal{S}_{2,d-1})}\left[|U_1|\right].$$

The density of $U_1$ is given by

$$f_{U_1}(x) \;=\; \frac{\Gamma\!\left(\frac{d}{2}\right)}{\sqrt{\pi}\,\Gamma\!\left(\frac{d-1}{2}\right)}\,(1-x^2)^{\frac{d-3}{2}}, \quad x\in[-1,1],$$

and the expectation can be computed as

$$\mathbb{E}\left[|U_1|\right] \;=\; \int_{-1}^{1} |x|\, f_{U_1}(x)\,\mathrm{d}x \;=\; 2\int_0^1 x\,\frac{\Gamma\!\left(\frac{d}{2}\right)}{\sqrt{\pi}\,\Gamma\!\left(\frac{d-1}{2}\right)}\,(1-x^2)^{\frac{d-3}{2}}\,\mathrm{d}x = \frac{2\Gamma\!\left(\frac{d}{2}\right)}{\sqrt{\pi}\,\Gamma\!\left(\frac{d-1}{2}\right)}\int_0^1 x\,(1-x^2)^{\frac{d-3}{2}}\,\mathrm{d}x$$

$$= \frac{\Gamma\!\left(\frac{d}{2}\right)}{\sqrt{\pi}\,\Gamma\!\left(\frac{d-1}{2}\right)}\int_0^1 (1-u)^{\frac{d-3}{2}}\,\mathrm{d}u \;=\; \frac{\Gamma\!\left(\frac{d}{2}\right)}{\sqrt{\pi}\,\Gamma\!\left(\frac{d-1}{2}\right)}\cdot\frac{1}{\frac{d-1}{2}} \;=\; \frac{2}{d-1}\,\frac{\Gamma\!\left(\frac{d}{2}\right)}{\sqrt{\pi}\,\Gamma\!\left(\frac{d-1}{2}\right)}.$$

Therefore, for large $d$, $\mathbb{E}\left[|U_1|\right] =_{d\to+\infty} \mathcal{O}(1/\sqrt{d})$. $\qquad\square$

## G. Laplace with Known Scale

In the following, $\theta$ denote the mean parameter of Laplace distribution with known scale $b$. We have $\mathcal{X}=\mathbb{R}$ and $k=d=1$. Let $\theta\in\Theta$ and $u\in\{\pm1\}$. It is direct to see that

$$\ell_\theta(x,y) = \log\frac{p_\theta(x)}{p_\theta(y)} = |y-\theta|/b - |x-\theta|/b = \frac{1}{b}\begin{cases} y-x & \text{if } \theta < \min\{x,y\} \\ x-y & \text{if } \theta > \max\{x,y\} \\ (2\theta-(x+y))\mathrm{sign}(x-y) & \text{if } \theta\in[\min\{x,y\},\max\{x,y\}] \end{cases},$$

and $\nabla_{\theta^\star}\ell_{\theta^\star}(x,y) = \begin{cases} 0 & \text{if } \theta^\star < \min\{x,y\} \text{ or } \theta^\star > \max\{x,y\} \\ \frac{2}{b}\mathrm{sign}(x-y) & \text{if } \theta^\star\in[\min\{x,y\},\max\{x,y\}] \end{cases}.$

Therefore, we have

$$\mathcal{G}_0(\theta^\star) = \left\{(x,y)\in\mathbb{R}^2 \mid ||y-\theta^\star|-|x-\theta^\star|| > 0\right\},$$
$$\mathcal{G}_1(\theta^\star) = \left\{(x,y)\in\mathbb{R}^2 \mid \theta^\star\in[\min\{x,y\},\max\{x,y\}]\right\},$$
$$\mathcal{G}_1(\theta^\star,u) = \left\{(x,y)\in\mathbb{R}^2 \mid \theta^\star\in[\min\{x,y\},\max\{x,y\}] \wedge u((x+y)/2-\theta^\star) > 0\right\},$$
$$\widetilde{\mathcal{D}}(\theta^\star,\theta) = \left\{(x,y)\in\mathbb{R}^2 \mid \theta^\star\in[\min\{x,y\},\max\{x,y\}] \wedge 0 < \mathrm{sign}(\theta-\theta^\star)((x+y)/2-\theta^\star) < |\theta-\theta^\star|\right\},$$
$$\forall(x,y)\in\mathcal{G}_1(\theta^\star,u), \quad V_{\theta^\star,u}(x,y) = u((x+y)/2-\theta^\star).$$

When $\theta^\star > \theta$, we have

$$\mathcal{D}(\theta^\star,\theta) = \{(x,y) \mid \{\theta^\star,\theta\}\subset[\min\{x,y\},\max\{x,y\}] \wedge \theta < (x+y)/2 < \theta^\star\}$$
$$\cup\{(x,y) \mid \theta < \min\{x,y\} \wedge \theta^\star\in((x+y)/2,\max\{x,y\}]\}$$
$$\cup\{(x,y) \mid \theta < \min\{x,y\} \wedge \theta^\star > \max\{x,y\}\}$$
$$\cup\{(x,y) \mid \theta^\star > \max\{x,y\} \wedge \theta\in[\min\{x,y\},(x+y)/2)\}.$$

When $\theta^\star < \theta$, we have

$$\mathcal{D}(\theta^\star,\theta) = \{(x,y) \mid \{\theta^\star,\theta\}\subset[\min\{x,y\},\max\{x,y\}] \wedge \theta^\star < (x+y)/2 < \theta\}$$
$$\cup\{(x,y) \mid \theta > \max\{x,y\} \wedge \theta^\star\in[\min\{x,y\},(x+y)/2)\}$$
$$\cup\{(x,y) \mid \theta^\star < \min\{x,y\} \wedge \theta > \max\{x,y\}\}$$
$$\cup\{(x,y) \mid \theta^\star < \min\{x,y\} \wedge \theta\in((x+y)/2,\max\{x,y\}]\}.$$

**Proof that** $\mathbb{P}_{p_{\theta^\star}^{\otimes 2}}(\mathcal{G}_1(\theta^\star)) > 0$. It is direct to see that $\dim(\mathcal{G}_0(\theta^\star)^\complement) < 2$. Given that $p_{\theta^\star}^{\otimes 2}$ is a continuous distribution on $(\mathbb{R})^2$, we obtain that $\mathbb{P}_{p_{\theta^\star}^{\otimes 2}}(\mathcal{G}_0(\theta^\star)) = 1$. Using the symmetry of the Laplace distribution around its mean, we have that

$$\mathbb{P}_{p_{\theta^\star}^{\otimes 2}}(\mathcal{G}_1(\theta^\star)) = \mathbb{P}_{p_{\theta^\star}^{\otimes 2}}((-\infty, \theta^\star) \times (\theta^\star, +\infty)) + \mathbb{P}_{p_{\theta^\star}^{\otimes 2}}((\theta^\star, +\infty) \times (-\infty, \theta^\star)) = 1/2\,.$$

**Condition in Lemma 3.1.** The condition of Lemma 3.1 is implied by Assumption 4.5, hence we refer to the proof of this result below. Therefore, we have $\mathcal{I}(q_{\theta^\star, h_{\mathrm{sto}}}) \succ \mathcal{I}(p_{\theta^\star}^{\otimes 2})$.

**Consistency of** $\mathrm{SP}_{\mathrm{det}}$. To study $\mathrm{SP}_{\mathrm{det}}$ for $\mathcal{F}_{\mathrm{Lap}, b}$, we use the change of variable $D = \theta^\star - X$ and $S = \theta^\star - Y$. For all $(D, S) \in \mathcal{G}_1(0)$, we have

$$\ell_{\theta^\star}(X, Y) = \frac{1}{b}(D + S)\mathrm{sign}(S - D)\,, \quad \nabla_{\theta^\star}\ell_{\theta^\star}(X, Y) = \frac{2}{b}\mathrm{sign}(S - D)\,, \quad \nabla_{\theta^\star}\log p_{\theta^\star}^{\otimes 2}(X, Y) = 0\,.$$

For all $(D, S) \notin \mathcal{G}_1(0)$, we have $\nabla_{\theta^\star}\ell_{\theta^\star}(X, Y) = 0$ and $\nabla_{\theta^\star}\log p_{\theta^\star}^{\otimes 2}(X, Y) \neq 0$. Let $M(D, S) = \mathbb{1}((D, S) \in \mathcal{G}_1(0))\,\sigma(-|D + S|/b)\mathrm{sign}(D + S)$. Then, $M(-D, -S) = -M(D, S)$ for all $(D, S) \in \mathbb{R}^2$. By integration of an odd function with respect to $0$ with a symmetric distribution around $0_2$, we obtain $\mathbb{E}_{(D,S) \sim \mathcal{N}(0_{2d}, I_{2d})}[M(D, S)] = 0$. Therefore, the condition (3) is satisfied and $\mathrm{SP}_{\mathrm{det}}$ is a consistent estimator.

**Asymptotic variance of** $\mathrm{SP}_{\mathrm{det}}$. Let $H_{\theta^\star}^{\mathrm{SP}_{\mathrm{det}}}$ and $R_{\theta^\star}^{\mathrm{SP}_{\mathrm{det}}}$ defined in Lemma 3.2. By definition of $\ell_\theta$, we obtain $\nabla_{\theta^\star}^2\ell_{\theta^\star} = 0$ and $H_{\theta^\star}^{\mathrm{SP}_{\mathrm{det}}} = 0$. Moreover, using the above formula, we have $\nabla_{\theta^\star}\ell_{\theta^\star}(X, Y)\nabla_{\theta^\star}\log p_{\theta^\star}^{\otimes 2}(X, Y) = 0$ for all $(D, S) \in \mathcal{G}_1(0)$, hence we obtain $R_{\theta^\star}^{\mathrm{SP}_{\mathrm{det}}} = 0$. The condition $\mathbb{P}_{p_{\theta^\star}^{\otimes 2}}(|\ell_{\theta^\star}\langle u, \nabla_{\theta^\star}\ell_{\theta^\star}\rangle| > 0) > 0$ for all $u \in \mathcal{S}_{d-1}$ is implied by Assumption 4.5, hence we refer to the proof of this result below. Using the sufficient condition derived in Appendix C.2, we have shown that $\mathrm{SP}_{\mathrm{det}}$ is asymptotically better than SP.

**Proof of Assumption 4.4.** Using that $\widetilde{\mathcal{D}}(\theta^\star, \theta) \subseteq \mathcal{G}_1(\theta^\star)$, we simply need to show that $\widetilde{\mathcal{D}}(\theta^\star, \theta) \subseteq \mathcal{G}_1(\theta^\star) \cap \mathcal{D}(\theta^\star, \theta)$. Let us consider the case $\theta^\star > \theta$. Then, we have

$$\begin{aligned}
\widetilde{\mathcal{D}}(\theta^\star, \theta) &= \left\{(x, y) \in \mathbb{R}^2 \mid \theta^\star \in [\min\{x, y\}, \max\{x, y\}] \wedge \theta < (x + y)/2 < \theta^\star\right\} \\
&= \{(x, y) \mid \{\theta^\star, \theta\} \subset [\min\{x, y\}, \max\{x, y\}] \wedge \theta < (x + y)/2 < \theta^\star\} \\
&\cup \{(x, y) \mid \theta < \min\{x, y\} \wedge \theta^\star \in ((x + y)/2, \max\{x, y\}]\} = \mathcal{G}_1(\theta^\star) \cap \mathcal{D}(\theta^\star, \theta)\,.
\end{aligned}$$

The same result follows when $\theta^\star < \theta$ by using the same argument. In summary, we have shown that $\widetilde{\mathcal{D}}(\theta^\star, \theta) = \mathcal{G}_1(\theta^\star) \cap \mathcal{D}(\theta^\star, \theta) \subseteq \mathcal{D}(\theta^\star, \theta)$.

**Proof of Assumption 4.5.** Using the symmetry of the Laplace distribution around its mean, we have $\mathbb{P}_{p_{\theta^\star}^{\otimes 2}}(\mathcal{G}_1(\theta^\star, u)) = \mathbb{P}_{p_{\theta^\star}^{\otimes 2}}(\mathcal{G}_1(\theta^\star, 1))$ for all $u \in \{\pm 1\}$. Then, by integrating for $x < y$, we obtain

$$\mathbb{P}_{p_{\theta^\star}^{\otimes 2}}(\mathcal{G}_1(\theta^\star, 1)) = \frac{1}{2b^2}\int_{x \in (-\infty, \theta^\star)} e^{x/b}\left(\int_{y \in (2\theta^\star - x, +\infty)} e^{-y/b}\mathrm{d}y\right)\mathrm{d}x = \frac{1}{2b}\int_{x \in (-\infty, \theta^\star)} e^{2x - 2\theta^\star/b}\mathrm{d}x = \frac{1}{4}\,.$$

**Proof of Assumption 4.7.** Let $\varepsilon > 0$. Using the symmetry of the Laplace distribution around its mean, we have $F_{\theta^\star, u}(\varepsilon) = F_{\theta^\star, 1}(\varepsilon)$ for all $u \in \{\pm 1\}$. Similarly as above, by integrating for $x < y$, we obtain that

$$\begin{aligned}
F_{\theta^\star, 1}(\varepsilon) &= \mathbb{P}_{(X,Y) \sim p_{\theta^\star}^{\otimes 2}}(0 < V_{\theta^\star, 1}(X, Y) \leq \varepsilon) \\
&= \frac{1}{2b^2}\int_{x \in (-\infty, \theta^\star)} e^{x/b}\left(\int_{y \in (2\theta^\star - x, 2\varepsilon + 2\theta^\star - x)} e^{-y/b}\mathrm{d}y\right)\mathrm{d}x \\
&= \frac{1}{2b}\left(\int_{x \in (-\infty, \theta^\star)} e^{(2x - 2\theta^\star)/b}\mathrm{d}x - \int_{x \in (-\infty, \theta^\star)} e^{(2x - 2\theta^\star - 2\varepsilon)/b}\mathrm{d}x\right) = \frac{1}{4}\left(1 - e^{-2\varepsilon/b}\right)\,.
\end{aligned}$$

Therefore, we have

$$F'_{\theta^\star,u}(x) = \frac{1}{2b}e^{-2\varepsilon/b} \quad , \quad F^{-1}_{\theta^\star,u}(x) = -\frac{b}{2}\log(1-4x) \quad \text{and} \quad (F^{-1}_{\theta^\star,u})''(x) = \frac{8b}{(1-4x)^2} \; .$$

Then, we obtain $F'_{\theta^\star,u}(0) = \frac{1}{2b}$ and we can take $x_{\theta^\star,u} = 1/8$ and $M_{\theta^\star,u} = 32b$.

**Proof of Assumption 4.2.** Let $\varepsilon = |\theta^\star - \theta|$ and $u = \text{sign}(\theta^\star - \theta)$. Using the above computation, we have

$$\mathbb{P}_{p_{\theta^\star}^{\otimes 2}}\left(\mathcal{D}(\theta^\star, \theta)\right) \geq \mathbb{P}_{p_{\theta^\star}^{\otimes 2}}\left(\widetilde{\mathcal{D}}(\theta^\star, \theta^\star + \varepsilon u)\right) \geq \mathbb{P}_{p_{\theta^\star}^{\otimes 2}}\left(\widetilde{\mathcal{D}}(\theta^\star, \theta^\star + \varepsilon u) \cap \mathcal{G}_1(\theta^\star, u)\right) = \mathbb{P}_{(X,Y)\sim p_{\theta^\star}^{\otimes 2}}(0 < V_{\theta^\star,u}(X,Y) < \varepsilon)$$

Using the above computation, we obtain that $\mathbb{P}_{(X,Y)\sim p_{\theta^\star}^{\otimes 2}}(0 < V_{\theta^\star,u}(X,Y) < \varepsilon) > 0$, hence $\mathbb{P}_{p_{\theta^\star}^{\otimes 2}}\left(\mathcal{D}(\theta^\star, \theta)\right) > 0$.

**Proof of Assumption 5.2.** Using that $f(x) = x^2 - 1 + (1+x)e^{-x}$ is positive on $\mathbb{R}_+$, we obtain

$$\text{H}^2(p_{\theta^\star}, p_\theta) = 1 - \left(1 + \frac{|\theta^\star - \theta|}{2b}\right)\exp\left(-\frac{|\theta^\star - \theta|}{2b}\right) \leq \frac{(\theta^\star - \theta)^2}{4b^2} \; .$$

First, we notice that $\dim\left(\mathcal{G}_0(\theta^\star)^\complement \triangle \mathcal{G}_0(\theta)^\complement\right) < 2$, hence we can show that

$$\int_{(x,y)\in\mathcal{G}_0(\theta^\star)^\complement \triangle \mathcal{G}_0(\theta)^\complement} \sqrt{p_{\theta^\star}(x)p_{\theta^\star}(y)p_\theta(x)p_\theta(y)}\text{d}x\text{d}y = 0 \; .$$

We consider the case $\theta^\star < \theta$ since $\theta^\star > \theta$ is done similarly as $\widetilde{\text{BC}}(\theta^\star, \theta) = \widetilde{\text{BC}}(\theta, \theta^\star)$. Let $\varepsilon = \theta - \theta^\star$. By integrating for $x < y$, we have

$$\begin{aligned}
\widetilde{\text{BC}}(\theta^\star, \theta) = {} & \frac{1}{2b^2}\int_x e^{x/b}\left(\int_y \mathbb{1}\left(x \leq \theta^\star < (x+y)/2 < \theta^\star + \varepsilon \leq y\right)e^{-y/b}\text{d}y\right)\text{d}x \\
& + \frac{e^{-(\varepsilon+\theta^\star)/b}}{2b^2}\int_x e^{x/b}\left(\int_y \mathbb{1}\left(y < \theta^\star + \varepsilon \wedge x \leq \theta^\star < (x+y)/2\right)\text{d}y\right)\text{d}x \\
& + \frac{e^{-\varepsilon/b}}{2b^2}\int_x\left(\int_y \mathbb{1}\left(\theta^\star < x < y < \theta^\star + \varepsilon\right)\text{d}y\right)\text{d}x \\
& + \frac{e^{-\theta^\star/b}}{2b^2}\int_y e^{-y/b}\left(\int_x \mathbb{1}\left(\theta^\star < x \wedge (x+y)/2 < \theta^\star + \varepsilon \leq y\right)\text{d}x\right)\text{d}y
\end{aligned}$$

Direct computation yields

$$\begin{aligned}
& \int_{x\in(\theta^\star-\varepsilon,\theta^\star)} e^{x/b}\left(\int_{y\in(2\theta^\star-x,\theta^\star+\varepsilon)} 1\text{d}y\right)\text{d}x = \int_{x\in(\theta^\star-\varepsilon,\theta^\star)} e^{x/b}(x+\varepsilon-\theta^\star)\text{d}x = e^{(\theta^\star-\varepsilon)/b}\int_{u\in(0,\varepsilon)} ue^{u/b}\text{d}u \; , \\
& \int_{u\in(0,\varepsilon)} ue^{u/b}\text{d}u = b\left(e^{\varepsilon/b}(\varepsilon-b)+b\right) \; , \\
& \int_x\left(\int_y \mathbb{1}\left(\theta^\star < x < y < \theta^\star + \varepsilon\right)\text{d}y\right)\text{d}x = \int_{x\in(\theta^\star,\theta^\star+\varepsilon)}(\theta^\star+\varepsilon-x)\text{d}x = \frac{\varepsilon^2}{2} \; , \\
& \int_{y\in(\theta^\star+\varepsilon,\theta^\star+2\varepsilon)} e^{-y/b}\left(\int_{x\in(\theta^\star,2\theta^\star+2\varepsilon-y)} 1\text{d}x\right)\text{d}y = \int_{y\in(\theta^\star+\varepsilon,\theta^\star+2\varepsilon)} e^{-y/b}(\theta^\star+2\varepsilon-y)\text{d}y \\
& \qquad = e^{-(\theta^\star+2\varepsilon)/b}\int_{u\in(0,\varepsilon)} ue^{u/b}\text{d}u \; .
\end{aligned}$$

Moreover, we have

$$
\int_x e^{x/b} \left( \int_y \mathbb{1}\left(x \le \theta^\star < (x+y)/2 < \theta^\star + \varepsilon \le y\right) e^{-y/b} \mathrm{d}y \right) \mathrm{d}x
$$

$$
= \int_{x \in (-\infty, \theta^\star - \varepsilon)} e^{x/b} \left( \int_{y \in (2\theta^\star - x, 2\theta^\star + 2\varepsilon - x)} e^{-y/b} \mathrm{d}y \right) \mathrm{d}x + \int_{x \in (\theta^\star - \varepsilon, \theta^\star)} e^{x/b} \left( \int_{y \in (\theta^\star + \varepsilon, 2\theta^\star + 2\varepsilon - x)} e^{-y/b} \mathrm{d}y \right) \mathrm{d}x
$$

$$
= b \int_{x \in (-\infty, \theta^\star - \varepsilon)} \left( e^{-(2\theta^\star - 2x)/b} - e^{-(2\theta^\star + 2\varepsilon - 2x)/b} \right) \mathrm{d}x + b \int_{x \in (\theta^\star - \varepsilon, \theta^\star)} \left( e^{-(\theta^\star + \varepsilon - x)/b} - e^{-(2\theta^\star + 2\varepsilon - 2x)/b} \right) \mathrm{d}x
$$

$$
= b \left( \frac{b}{2} e^{-2\varepsilon/b} - \frac{b}{2} e^{-4\varepsilon/b} + b \left( e^{-\varepsilon/b} - e^{-2\varepsilon/b} \right) + \frac{b}{2} \left( e^{-4\varepsilon/b} - e^{-2\varepsilon/b} \right) \right) = b^2 \left( e^{-\varepsilon/b} - e^{-2\varepsilon/b} \right)
$$

Therefore, we have

$$
\widetilde{\mathrm{BC}}(\theta^\star, \theta^\star + \varepsilon) = \frac{1}{2} \left( e^{-\varepsilon/b} - e^{-2\varepsilon/b} \right) + \frac{1}{2b} \left( e^{-2\varepsilon/b} + e^{-2(\theta^\star + \varepsilon)/b} \right) \left( e^{\varepsilon/b}(\varepsilon - b) + b \right) + e^{-\varepsilon/b} \frac{\varepsilon^2}{4b^2}
$$

$$
= \frac{1}{2} \left( e^{-\varepsilon/b} - e^{-2\varepsilon/b} \right) + \frac{1}{2} \left( e^{-2\varepsilon/b} + e^{-2(\theta^\star + \varepsilon)/b} \right) \left( \frac{\varepsilon}{b} e^{\varepsilon/b} - e^{\varepsilon/b} + 1 \right) + e^{-\varepsilon/b} \frac{\varepsilon^2}{4b^2}
$$

$$
= \frac{1}{2} \left( e^{-2(\theta^\star + \varepsilon)/b} - e^{-(2\theta^\star + \varepsilon)/b} \right) + \frac{1}{2} \left( e^{-\varepsilon/b} + e^{-(2\theta^\star + \varepsilon)/b} \right) \frac{\varepsilon}{b} + e^{-\varepsilon/b} \frac{\varepsilon^2}{4b^2}
$$

$$
= \frac{1}{2} e^{-\varepsilon/b} \left( e^{-2\theta^\star/b}(e^{-\varepsilon/b} - 1 + \varepsilon/b) + \frac{\varepsilon}{b} + \frac{\varepsilon^2}{2b^2} \right) .
$$

Then, we can conclude that

$$
\widetilde{\mathrm{BC}}(\theta^\star, \theta^\star - \varepsilon) = \widetilde{\mathrm{BC}}(\theta^\star - \varepsilon, \theta^\star) = \frac{1}{2} e^{-\varepsilon/b} \left( e^{-2(\theta^\star - \varepsilon)/b}(e^{-\varepsilon/b} - 1 + \varepsilon/b) + \frac{\varepsilon}{b} + \frac{\varepsilon^2}{2b^2} \right) .
$$

Using that $f(x) = 1 - x + x^2/2 - e^{-x}$ is positive on $\mathbb{R}_+$, we obtain

$$
\widetilde{\mathrm{BC}}(\theta^\star, \theta) \le \frac{|\theta^\star - \theta|}{2b} \left( 1 + \frac{|\theta^\star - \theta|}{2b} \left( 1 + e^{-2 \min\{\theta^\star, \theta\}/b} \right) \right) .
$$

# H. Supplementary Experiments

Using the same empirical setup as in Section 6, we conduct additional experiments to support our theoretical claims for other distributions (Appendix H.1), other estimators for Gaussian distributions based on $\mathcal{C}_n$ (Appendix H.2), other convex surrogates of the 0-1 loss (Appendix H.3) or normalized/regularized versions of the logistic loss (Appendix H.4).

**Reproducibility.** Code for reproducing our empirical results is available at https://github.com/tml-epfl/learning-parametric-distributions-from-samples-and-preferences. Our code is implemented in Julia (Bezanson et al., 2017), version 1.11.5. The plots are generated with StatsPlots. The optimization problems defining some of our estimators are solved numerically with JuMP (Lubin et al., 2023), by using the Ipopt (Wächter & Biegler, 2006) and HiGHS (Huangfu & Hall, 2018) solvers. Other dependencies are listed in the Readme.md that provides detailed julia instructions to reproduce our experiments, as well as a script.sh to run them all at once. Our experiments are conducted on 12 Intel(R) Core(TM) Ultra 7 165U 4.9GHz CPU.

**Gaussian distribution with known variance.** For $\mathcal{F}_{\mathcal{N},1}$, the $\mathrm{SP}_{\mathrm{det}}$ and SP estimators are computed with the Ipopt solver. For $\mathcal{F}_{\mathcal{N}, I_d}$, the $\mathrm{SP}_{\mathrm{det}}$, SP, DP and WE estimators are computed with the Ipopt solver, and the AE estimator uses the HiGHS solver.

## H.1. Accelerated Rates for Other Distributions

### H.1.1. LAPLACE DISTRIBUTION WITH KNOWN SCALE

**Estimators.** For $\mathcal{F}_{\mathrm{Lap},1}$ (Appendix G), we have

$$
\widehat{\theta}_n^{\mathrm{SO}} = \mathrm{median}\left(\{X_i\}_{i \in [n]} \cup \{Y_i\}_{i \in [n]}\right) \quad \text{and} \quad \mathcal{C}_n = \left\{\theta \mid \forall i \in [n], \, Z_i(|Y_i - \theta| - |X_i - \theta|) \ge 0\right\} .
$$

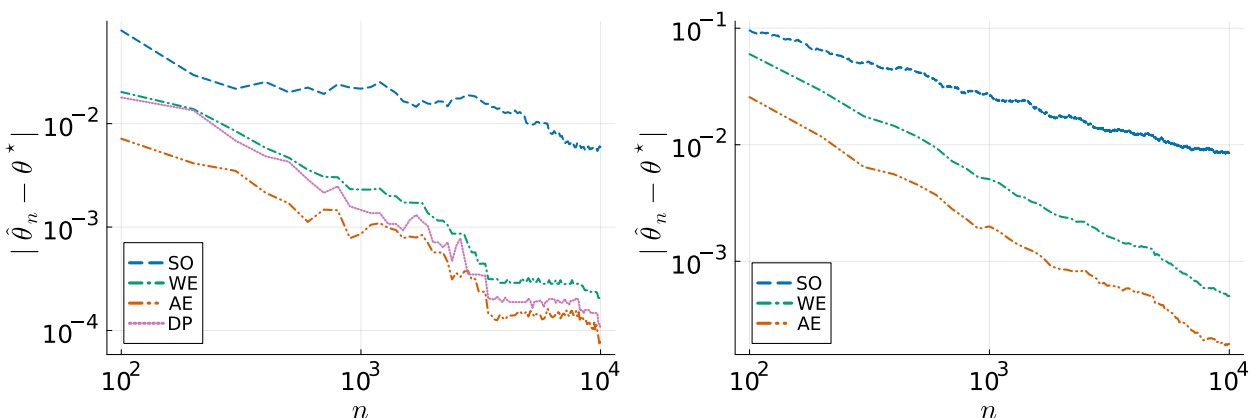

*Figure 4.* Estimation errors for (a) $\text{Lap}(\theta^\star, 1)$ where $\theta^\star \sim \mathcal{U}([1, 2])$ with $N_{\text{runs}} = 10$ and (b) $\text{Rayleigh}(\sqrt{\theta^\star})$ where $\theta^\star \sim \mathcal{U}([1, 2])$ with $N_{\text{runs}} = 10^2$.

The estimators based on $\mathcal{C}_n$ are $\widehat{\theta}_n^{\text{AE}} \in \mathcal{C}_n$, $\widehat{\theta}_n^{\text{WE}} := \arg\max_{\theta \in \mathcal{C}_n} |\theta - \theta^\star|$ and

$$\widehat{\theta}_n^{\text{DP}} = \arg\max_{\theta \in \mathcal{C}_n} \sum_{i \in [n]} \left( |Y_i - \theta| + |X_i - \theta| \right) .$$

Those three estimators are computed with the `Ipopt` solver.

**Experiments.** Figure 4(a) confirms empirically the difference in estimation rate between the M-estimators (SO MLE)—obtaining $\mathcal{O}(1/\sqrt{n})$—and our estimators based on $\mathcal{C}_n$—achieving $\mathcal{O}(1/n)$. Moreover, AE and WE perform on par with DP MLE.

### H.1.2. RAYLEIGH DISTRIBUTION

Let $\sigma > 0$ be the scale parameter characterizing a Rayleigh distribution. In the following, let $\theta = -\frac{1}{2\sigma^2} < 0$ denote the natural parameter of a Rayleigh distribution. We have $\Theta \subseteq \mathbb{R}_-^\star$, $\mathcal{X} = \mathbb{R}_+$ and $k = d = 1$. The probability density function is defined as

$$\forall x \in \mathbb{R}_+, \quad p_\theta(x) = \exp\left(x^2\theta + \log(x) + \log(2\theta)\right) .$$

Let $\theta \in \Theta$ and $u \in \{\pm 1\}$. It is direct to see that, for all $(x, y) \in \mathbb{R}_+^2$,

$$\ell_\theta(x, y) = \log\frac{p_\theta(x)}{p_\theta(y)} = (x^2 - y^2)\theta + \log(x/y) \quad \text{and} \quad \frac{\mathrm{d}\ell_{\theta^\star}}{\mathrm{d}\theta^\star}(x, y) = x^2 - y^2 = (x - y)(x + y) .$$

Therefore, we have

$$\mathcal{G}_0(\theta^\star) = \{(x, y) \in \mathbb{R}_+^2 \mid |(x^2 - y^2)\theta^\star + \log(x/y)| > 0\} ,$$
$$\mathcal{G}_1(\theta^\star) = \{(x, y) \in \mathbb{R}_+^2 \mid |x - y| > 0\} ,$$
$$\mathcal{G}_1(\theta^\star, u) = \{(x, y) \in \mathbb{R}_+^2 \mid u((x^2 - y^2)^2\theta^\star + (x^2 - y^2)\log(x/y)) < 0\} ,$$
$$\mathcal{D}(\theta^\star, \theta) = \{(x, y) \in \mathbb{R}_+^2 \mid ((x^2 - y^2)\theta^\star + \log(x/y))^2 + (x^2 - y^2)(\theta - \theta^\star)\left((x^2 - y^2)\theta^\star + \log(x/y)\right)\} ,$$
$$\forall (x, y) \in \mathcal{G}_1(\theta^\star, u), \quad V_{\theta^\star, u}(x, y) = -u\left(\theta^\star + \frac{1}{x + y}\frac{\log(x) - \log(y)}{x - y}\right) .$$

**Proof that** $\mathbb{P}_{p_{\theta^\star}^{\otimes 2}}(\mathcal{G}_1(\theta^\star)) > 0$. It is direct to see that $\dim(\mathcal{G}_0(\theta^\star)^\complement) < 2$ and $\dim(\mathcal{G}_0(\theta^\star) \setminus \mathcal{G}_1(\theta^\star)) < 2$. Given that $p_{\theta^\star}^{\otimes 2}$ is a continuous distribution on $(\mathbb{R}_+)^2$, we obtain that $\mathbb{P}_{p_{\theta^\star}^{\otimes 2}}(\mathcal{G}_0(\theta^\star)) = \mathbb{P}_{p_{\theta^\star}^{\otimes 2}}(\mathcal{G}_1(\theta^\star)) = 1$.

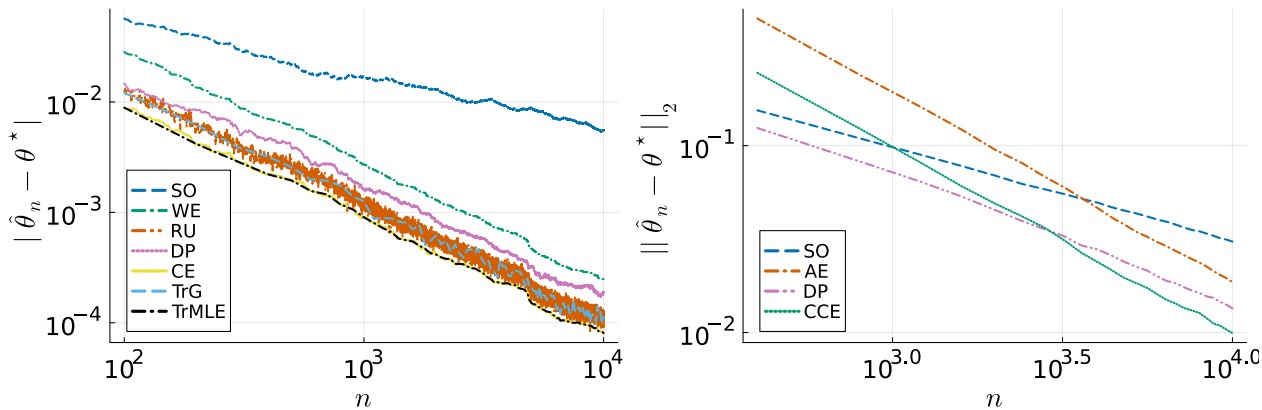

Figure 5. Estimation errors for $\mathcal{N}(\theta^\star, I_d)$ where $\theta^\star \sim \mathcal{U}([1,2]^d)$ with $N_{\text{runs}} = 10^2$ for (a) $d = 1$ and (b) $d = 20$.

**Proof of Assumption 4.4 and 4.5.** Since $\ell_\theta(x,y) = (x^2 - y^2)\theta + \log(x/y)$ is linear in $\theta$, we have $\mathcal{D}(\theta^\star, \theta) = \widetilde{\mathcal{D}}(\theta^\star, \theta)$. Let $(X, Y) \sim p_{\theta^\star}^{\otimes 2}$. Then, we have

$$\mathbb{P}_{p_{\theta^\star}^{\otimes 2}}(\mathcal{G}_1(\theta^\star, 1)) = \mathbb{P}_{(X,Y) \sim p_{\theta^\star}^{\otimes 2}}\left(\theta^\star < \frac{1}{X^2}\frac{\log(Y/X)}{1 - (Y/X)^2}\right) > 0\,,$$

$$\mathbb{P}_{p_{\theta^\star}^{\otimes 2}}(\mathcal{G}_0(\theta^\star, -1)) = \mathbb{P}_{(X,Y) \sim p_{\theta^\star}^{\otimes 2}}\left(\theta^\star > \frac{1}{X^2}\frac{\log(Y/X)}{1 - (Y/X)^2}\right) > 0\,.$$

**Estimators.** We have

$$\widehat{\theta}_n^{\text{SO}} = \frac{1}{4n}\sum_{i \in [n]}(X_i^2 + Y_i^2) \quad \text{and} \quad \mathcal{C}_n = \{\theta \mid \forall i \in [n],\ Z_i((X_i^2 - Y_i^2)\theta + \log(X_i/Y_i)) \geq 0\}\,.$$

The estimators based on $\mathcal{C}_n$ are $\widehat{\theta}_n^{\text{AE}} \in \mathcal{C}_n$, $\widehat{\theta}_n^{\text{WE}} := \arg\max_{\theta \in \mathcal{C}_n} |\theta - \theta^\star|$. Those two estimators are computed with the `Ipopt` solver.

**Experiments.** Figure 4(b) confirms empirically the difference in estimation rate between the M-estimators (SO MLE)—obtaining $\mathcal{O}(1/\sqrt{n})$—and our estimators based on $\mathcal{C}_n$—achieving $\mathcal{O}(1/n)$. Moreover, AE and WE perform similarly.

### H.2. Other Estimators for Gaussian Distributions

To better understand the surprising performance of the RU estimator, we consider other estimators that disentangle the effect of RU's randomness versus its mean behavior.

**Univariate Gaussian.** The center estimator (CE) returns the center of the interval $\mathcal{C}_n$. The truncated Gaussian estimator (TrG) returns a realization from a Gaussian distribution with mean CE and variance $4/n$, which is truncated to $\mathcal{C}_n$. The truncated MLE (TrMLE) returns the average of the observations $(\{X_i\}_{i \in [n]} \cup \{Y_i\}_{i \in [n]}) \cap \mathcal{C}_n$.

Figure 5(a) reveals that TrG performs on par with RU, yet CE and TrMLE outperform both TrG and RU. This suggests that being far away from the boundary of $\mathcal{C}_n$ improves performance compared to DP that lies on the boundary of $\mathcal{C}_n$ (as observed empirically). Moreover, randomization on $\mathcal{C}_n$ worsens performance compared to CE.

Using the derivation in the introduction on univariate Gaussian, it is coherent that CE improves on DP by a multiplicative constant: the average of those two (non-independent) random variables decreases faster. Formally, this could be proven by refining the proof of Lemma 4.6 to account for the property that $n = N_{\theta^\star, -1} + N_{\theta^\star, 1}$.

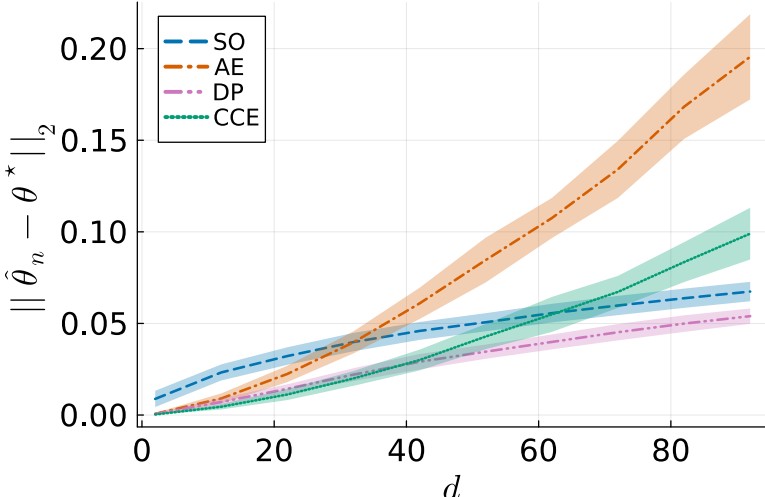

*Figure 6.* Estimation errors as a function of $d$ with $\mathcal{N}(\theta^\star, I_d)$ where $\theta^\star \sim \mathcal{U}([1,2]^d)$, for $n = 10^4$ and $N_{\text{runs}} = 10^2$.

**Multivariate Gaussian.** For $d > 1$, multiple centers exist. We use the Chebyshev center estimator (CCE) of $\mathcal{C}_n$.

Figures 5(b) and 6 shows that CCE outperforms AE by a constant margin. It only outperforms DP in the regime of large $n$ compared to $d$ and performs worse than SO MLE for small $n$. Geometrically, for small $n$ and large $d$, we conjecture that the random polytope $\mathcal{C}_n$ is more likely to be "spiky" along some directions. Due to those distant vertices, the center would become a worse estimator than DP, since the "average" is intuitively less robust to outliers. In contrast, DP MLE dominates SO MLE statistically (Lemma 4.1), hence it achieves rate $O(\sqrt{d/n})$ when $n$ is small compared to $d$.

### H.3. Estimators Based on Convex Surrogate of the 0-1 Loss

While DP MLE minimizes an objective that minimizes the 0-1 loss, SP MLE minimizes an objective involving the logistic loss $f_{\text{Log}}(x) = \log(1 + \exp(-x))$. As in Tang et al. (2024b), we can generalize this approach to $f$ any convex surrogate of the 0-1 loss, see Figure 7(a). For example, we consider the Hinge loss (Hin), i.e., $f_{\text{Hin}}(x) := \max\{0, 1 - x\}$, the square loss (Squ), i.e., $f_{\text{Squ}}(x) := (1 - x)^2$, the truncated square loss (TrS), i.e., $f_{\text{TrS}}(x) := \max\{0, 1 - x\}^2$, the Savage loss (Sav), i.e., $f_{\text{Sav}}(x) := (1 + \exp(x))^{-2}$, and the exponential loss (Exp), i.e., $f_{\text{Exp}}(x) := \exp(-x)$.

Given $(X_i, Y_i, Z_i)_{i \in [n]} \sim q_{\theta^\star, h_{\text{det}}}^{\otimes[n]}$ and a loss $f$, we consider the estimator

$$\widehat{\theta}_n^f \in \arg\min_{\theta \in \Theta} \left\{ L_n^{\text{SO}}(\theta) + \sum_{i \in [n]} f(Z_i \ell_\theta(X_i, Y_i)) \right\} .$$

All those estimators are computed with the `Ipopt` solver.

Figure 7(b) shows that all estimators perform on par with SP MLE, i.e., the one based on the logistic loss.

### H.4. Impact of Normalization and Regularization

The estimator defined in Appendix H.3 can be further generalized by introducing a regularization parameter $\lambda \geq 0$ and a normalization parameter $\beta > 0$, see, e.g., Gorbatovski et al. (2025). Given $(X_i, Y_i, Z_i)_{i \in [n]} \sim q_{\theta^\star, h_{\text{det}}}^{\otimes[n]}$, a loss $f$ and regularization/normalization $(\lambda, \beta)$, we consider the estimator

$$\widehat{\theta}_n^{f, \lambda, \beta} \in \arg\min_{\theta \in \Theta} \left\{ L_n^{\text{SO}}(\theta) + \lambda \sum_{i \in [n]} f(\beta Z_i \ell_\theta(X_i, Y_i)) \right\} .$$

While similar modifications could be made for other losses, we focus on the logistic loss $f_{\text{Log}}(x) = \log(1 + \exp(-x))$. In particular, we recover SP MLE by taking $\lambda = \beta = 1$.

Figures 8(a) and (b) showcase the "mild" impact of normalization and regularization.

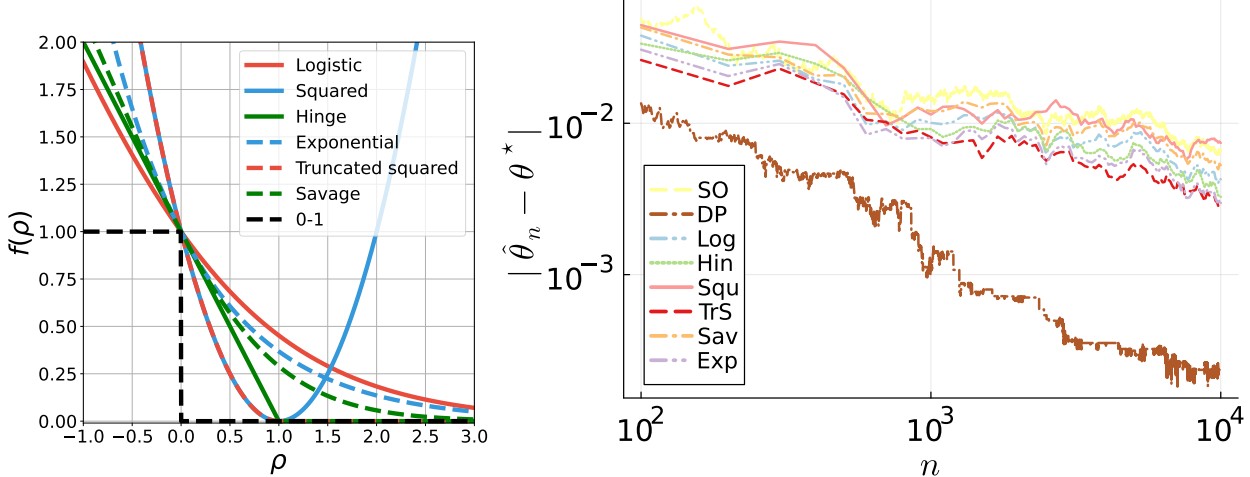

*Figure 7.* (a) Figure 2 in Tang et al. (2024b): notable examples of binary classification loss functions. (b) Estimation errors when minimizing the empirical losses for $\mathcal{N}(\theta^\star, 1)$ where $\theta^\star \sim \mathcal{U}([1, 2])$ with $N_{\text{runs}} = 10$.

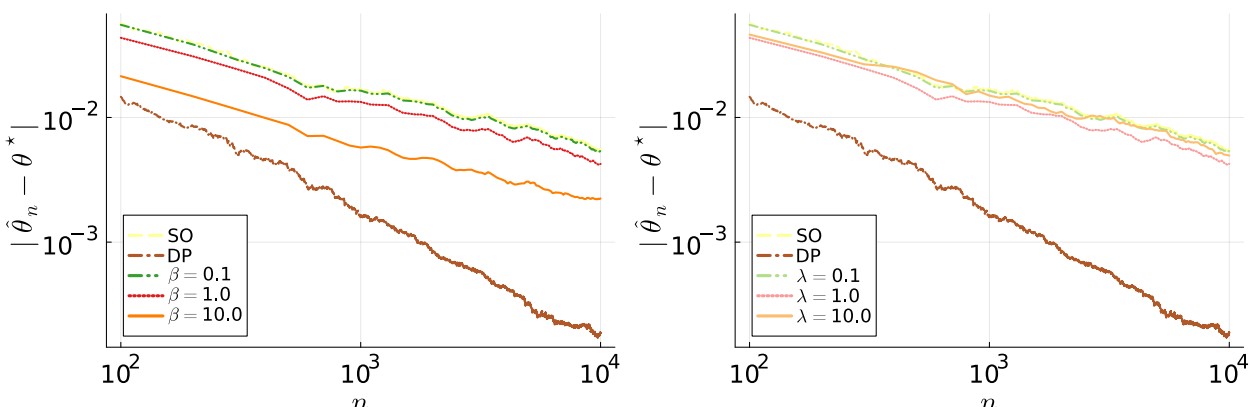

*Figure 8.* Estimation errors when minimizing the empirical losses for $\mathcal{N}(\theta^\star, 1)$ where $\theta^\star \sim \mathcal{U}([1, 2])$ with $N_{\text{runs}} = 10^2$ when (a) normalizing by $\beta$ with regularization $\lambda = 1$ and (b) regularizing by $\lambda$ with normalization $\beta = 1$.

