# OpenReview forum: "Learning Parametric Distributions from Samples and Preferences"
_ICML.cc/2025/Conference — ICML 2025 spotlightposter_

### Official Review · Reviewer_yquN · 2025-03-07

**Overall Recommendation:** 4

**Summary:**

This paper studies the conditions under which preference feedback improves parameter estimation. The authors show that preference-based estimator can achieve a better assympototic variance than sample-only estimators. When incorporated with hard constraints with deterministic preference, the authors prove an estimation error of $\mathcal{O}(1/n)$, improving upon traditional rate $\mathcal{O}(1/\sqrt{n})$, under some restrictive assumptions. They also develop a matching lower bound.

**Claims And Evidence:**

Yes

**Essential References Not Discussed:**

No

**Experimental Designs Or Analyses:**

Yes, in Section 6

**Methods And Evaluation Criteria:**

Yes

**Other Comments Or Suggestions:**

see Strengths And Weaknesses part above

**Other Strengths And Weaknesses:**

## Strengths

The paper studies a very interesting problem, i.e. how does preference labels improve statistical estimation. The theoretical results are solid and also surprising overall, providing important insights on the benefits of preference.

## Weaknesses

As mentioned by authors, the assumptions are quite restrictive and are only verified under simple setup. This is not a big issue, though.

**Questions For Authors:**

### 1.
Do the results in Section 4 rely on the specific reward model $r_\theta$? Does it have to be the log likelihood $\log p_\theta$? If not, can the authors provide some examples satisfying all the assumptions while supporting general reward functions?

### 2.
Can the authors provide more high-level intuitions behind the accelerated rate of $\mathcal{O}(1/n)$? For examples, when the reward model is just log likelihood, the deterministic preference label only provides additional information on the magnitude relationship between $\log p(x),\log p(y)$. Why is this sufficient to improve the estimation error?

### 3.
Is it possible to consider misspecification case, i.e., $\theta_*\not\in \Theta$? In this case, $\hat{\theta}$ should converge to an optimal estimator in $\Theta$. Would similar acceleration effects hold?

**Relation To Broader Scientific Literature:**

It is generally related to machine learning community

**Theoretical Claims:**

No

---

> ### Author Rebuttal · Authors · 2025-03-28
>
> We thank Reviewer yquN for the time spent and the positive feedback. We address the reviewer’s questions below.
>
> **1. Reward models**
> Except for Theorem 4.3, all the derivations in Section 4 hold for general (hence reward-based) preference models provided Assumptions 4.2,4.4,4.5 and 4.7 hold. Characterizing the expressivity of parametric rewards satisfying those assumptions is interesting, yet challenging. We provide two positive and one negative examples.
> - Positive: monotonic reward. Suppose that $\tilde \ell_{\theta}(x,y) = f(p_{\theta}(x)) - f(p_{\theta}(x))$ where $f$ is increasing on $[0,1]$. Since $sign(\tilde \ell_{\theta}) = sign(\ell_{\theta})$, hence the parameters with zero classification loss and our estimators are the same. Therefore, our results hold for this class of rewards when our assumptions hold for the log-likelihood reward. When $f$ is decreasing, the preferences are “reversed”, and similar arguments can be made.
> This example includes (1) normalization by a multiplicative constant (e.g., temperature $\beta$) and (2) the odds-ratio reward-based preference based on $f(x) = \log(x/(1-x))$ and defined by Hong et al. (2024, ORPO: Monolithic Preference Optimization without Reference Model).
> - Positive: margin with Gaussian. Suppose that $\tilde \ell_{\theta} = \ell_{\theta} + c$ where $c$ is a constant and $\ell_{\theta}$ is the Gaussian log-likelihood preference. By extending our computations from Appendix E, Assumptions 4.2, 4.4, 4.5, and 4.7 hold with $c$-dependent positive constants.
> Margins are used by Meng et al. (2024, SimPO: Simple Preference Optimization with a Reference-Free Reward) and IPO from Azar et al. (2023, A General Theoretical Paradigm to Understand Learning from Human Preferences).
> - Negative: reference model with Gaussian. Suppose that $\tilde \ell_{\theta} = \ell_{\theta} - \ell_{\theta_0}$ where $\theta_0$ is known and $\ell_{\theta}$ is the Gaussian log-likelihood preference. Since $\tilde \ell_{\theta}(x,y) = \langle x-y, \theta - \theta_0 \rangle$ and $\nabla_{\theta} \tilde \ell_{\theta}(x,y) = x-y$, Assumption 4.5 is violated for $u=\theta^\star - \theta_0$. Not all direct alignment algorithms rely on a reference model (see SimPO or ORPO).
>
> **2. Accelerated rate**
> Accelerated rates arise when accumulating random variables having a positive density at a specific point through a minimum (or maximum) operator.
> - When estimating the location parameter $\theta$ of a uniform distribution over $[\theta,\theta+1]$, the optimal estimator achieving the accelerated rate is the minimum of uniform observations whose density is positive at $\theta$.
> - For deterministic preferences with log-likelihood rewards, we observe the true ordering between likelihoods. This enforces a hard constraint on the admissible parameters, which can be expressed with a minimum operator. More precisely, the maximal deviation $R_{n,u}$ along direction $u$ is upper bounded by the minimum of positive random variables whose density is positive at zero under Assumption 4.7. With high probability, this min operator is upper bounded by $O(1/n)$. The proof combines Lemma 4.6 and an upper bound on the inverse of the cdf based on a Taylor expansion around $0$.
>
> **3. Misspecification**
> Under misspecification, the deterministic preferences might not provide separability within $\Theta$ since $\theta^\star \notin \Theta$. Then, DP MLE should be defined as SP MLE by using the 0-1 loss $1(u < 0)$ instead of the logistic loss $-\log \sigma(u)$. This combines a cross-entropy loss and a classification 0-1 loss, reweighted by a regularization $\lambda > 0$. This objective is reminiscent of single-stage alignment procedures such as ORPO and ASFT, see Gorbatovski et al. (2025, The Differences Between Direct Alignment Algorithms are a Blur).
> Without separability, computing DP MLE can be NP-hard. Under sufficient regularity, DP MLE converges to $\theta_{0} \in argmin_{\theta \in \Theta} KL(\theta^\star,\theta) + \lambda m(\theta)$ where $m$ as in line 270, where $\theta_0 \ne \theta^\star$. This minimization is challenging, as $\theta \to m(\theta)$ might not be convex. Deriving a tractable ELBO method for this optimization is an interesting direction to obtain tractable and robust estimators.
> As $\theta_0$ lies in the boundary of $\Theta$, we should control the maximal deviation wrt to $\theta_0$ for directions that point towards the interior of $\Theta$ to prove an accelerated rate. While some elements of our analysis might be salvaged, we believe that finer technical arguments should be derived to capture this interesting setting.
>
> **Restrictive assumptions**
> See the answer to Reviewer vgrP for a detailed discussion on how to weaken them to local conditions.

---

> > ### Comment · Reviewer_yquN · 2025-04-02
> >
> > Thanks for the authors' feedback. I keep my recommendation for acceptance.

---

### Official Review · Reviewer_Gth4 · 2025-03-13

**Overall Recommendation:** 4

**Summary:**

This paper studies when adding preference feedback can boost the parameter estimation in the cases of Gaussian and Laplace distributions. The results are mainly theoretical, containing three parts: (1) For M-estimators, adding an additional ``preference'' term related to the logarithm of probability helps reduce the asymptotic covariance; (2) For estimators based on hard preference constraints, the error converges at a rate of $\mathcal{O}(1/n)$ with high probability; (3) This rate is mini-max optimal up to dimension and problem-dependent constants, using Assaud's Lemma.

**Claims And Evidence:**

Most of the results are theoretical and supported by proofs. The assumptions are satisfied by the Gaussian or the Laplace distributions.

**Essential References Not Discussed:**

No (as far as I know).

**Experimental Designs Or Analyses:**

I checked the experiment results. As I mentioned before, this paper is theoretical. Yet there are still some minor issues.
1. It seems a little weird that any randomized estimator in $\mathcal{C}_n$ (RU) outperforms the one that maximizes the log-likelihood in $\mathcal{C}_n$ (DP) in Figure 1(a).
2. The legend doesn't match the figure's line style.

**Methods And Evaluation Criteria:**

The estimators in this paper are mostly theoretical. SO and SP are typical M-estimators. AE and DP require (1) solving a feasibility problem, which can be NP-hard for general cases; and (2) the hard preference assumption (analogous to the feasibility condition). I don't think this could be the real case.

**Other Comments Or Suggestions:**

None.

**Other Strengths And Weaknesses:**

None.

**Questions For Authors:**

None.

**Relation To Broader Scientific Literature:**

This paper provides a new perspective in analyzing the role of preference. This paper's case is not the same as the human preference alignment (e.g. RLHF): the paper is studying an estimator ``plus'' some preference data as an additional source of information, while RLHF or DPO is trying to learn something from only the preference data, implying that the paper may be of limited value to the LLM literature. However, this paper still provides some interesting observations, which might be of interest to the community of statistics.

**Theoretical Claims:**

I checked the proof sketch. It makes sense to me.

---

> ### Author Rebuttal · Authors · 2025-03-28
>
> We thank Reviewer Gth4 for the time spent and the encouraging feedback. We address the reviewer’s questions below.
>
> **Iterative human preference alignment**
> We investigate the case where pairs of observations and their preferences are tied together, which includes the log-likelihood ratio as preference. We detail the connections with iterative human preference alignment.
> - Many human preference alignment methods build on the Bradley-Terry (BT) model for preference, based on rewards. Direct alignment algorithms use variants of the log-likelihood to define the implicit reward of a policy. Choosing $\ell_{\theta}(x,y) = \log p_{\theta}(x) - \log p_{\theta}(y)$ coincides with the optimal policy for maximum entropy RL (see, e.g., Swamy et al, 2025, All Roads Lead to Likelihood: The Value of Reinforcement Learning in Fine-Tuning).
> - For offline preference data, the assumption $(X,Y) \sim p_{\theta^\star}^{\otimes 2}$ is unrealistic as $\ell_{\theta^\star}$ is collected from a fixed data set of pairs of observations. Recent LLMs are built on iterative alignment procedures. At stage $N$, the model $p_{\theta_N}$ is trained based on the preference data for generations by the previous model, i.e., $(X,Y) \sim p_{\theta_{N-1}}^{\otimes 2}$. Under the realizability assumption and without mode collapse, this self-refinement paradigm converges towards the true model $p_{\theta^\star}$. Our setting characterizes the limiting behavior of this iterative process, i.e., preference based on $\ell_{\theta^\star}$ for observations from $p_{\theta^\star}$.
>
> **1. RU versus DP**
> Figure 1(a) provides evidence suggesting that the randomized estimator (RU) and the worst-case estimator (WE) perform on par with DP MLE: RU is slightly better than DP, itself slightly better than WE. Figures 1(b) and (2) highlight that DP outperforms WE and AE for larger dimensions, where the gap increases when $d$ is nonnegligible compared to $n$. Therefore, only DP obtains the best-of-both world estimation error rate. For large $d$, implementing RU is challenging. We conjecture it suffers from the same limitation as AE. This is supported by additional experiments on new estimators using the setting of Section 6, see anonymous plots at https://anonymous.4open.science/r/ICML25SuppExp .
> - Figure 1(a) extended. The center estimator (CE) returns the center of the interval $\mathcal C_n$. The truncated Gaussian estimator (TrG) returns a realization from a Gaussian distribution with mean CE and variance $4/n$, which is truncated to $\mathcal C_n$. TrG performs on par with RU. CE outperforms both TrG and RU. This suggests that being far away from the boundary of $\mathcal C_n$ improves performance compared to DP that lies on the boundary of $\mathcal C_n$ as observed empirically. Moreover, randomization on $\mathcal C_n$ worsens performance compared to CE.
> Using the derivation in lines 55-63, it is coherent that CE improves on DP by a multiplicative constant: the average of those two (non-independent) random variables decreases faster. This can be proved by refining the proof of Lemma 4.6 to account that $n = N_{\theta^\star,-1} + N_{\theta^\star,1}$ (defined in Line 658).
> - Figures 1(b) and 2 extended. For $d>1$, multiple centers exist and we use the Chebyshev center estimator (CCE) of $\mathcal C_n$. While CCE outperforms AE by a constant margin, CCE only outperforms DP in the regime of large $n$ compared to $d$. It performs worse than SO for small $n$. Geometrically, for small $n$ and large $d$, the random polytope $\mathcal C_n$ is more likely to be “spiky” along some directions. Due to those distant vertices, the center becomes a worse estimator than DP, since the “average” is intuitively less robust to outliers. In contrast, DP dominates SO statistically (Lemma 4.1), hence it achieves rate $O(\sqrt{d/n})$ when $n$ is small compared to $d$.
>
> **2. Line style**
> The dashed line is shorter than the solid line, see SP (sto) and SO, yet others are not distinguishable. We will correct this.

---

### Official Review · Reviewer_vgrP · 2025-03-23

**Overall Recommendation:** 3

**Summary:**

The paper provides a set of estimators and conditions to improve the estimation error in learning the parameters of continuous parametric distributions when additional preference feedback is available. More concretely, the question is the following: For a continuous parametric distribution $p\_\theta$ with i.i.d. samples $\{(x\_i, y\_i)\}_i$ and a known reward function $r\_\theta$, how/when does including noisy/deterministic preferences $z\_i \propto r\_\theta(x\_i) - r\_\theta(y\_i)$ improve the estimation error of the parameter $\theta$?

To answer the above question, the authors first leverage the asymptotic theory of M-estimators, showing that a maximum-likelihood estimator (MLE) that takes preference data into account has the same standard error rate as sample-only estimators, i.e., $\Theta(\frac{1}{\sqrt{n}})$, while achieving a potentially improved asymptotic variance for noisy preferences and a further improved variance for deterministic preferences.

For deterministic preferences, they take a further step and provide another estimator: an MLE with the hard constraints given by preferences. They then make several assumptions on $p_\theta$ and $r_\theta$ to show that this new estimator can achieve an accelerated error rate of $\mathcal{O}(\frac{1}{n})$ compared to the standard $\Theta(\frac{1}{\sqrt{n}})$. In particular, they show that Normal and Laplace distributions with log-probability reward functions satisfy these assumptions.

Finally, they prove that the rate of $\mathcal{O}(\frac{1}{n})$ is minimax optimal up to problem-specific dimensions and logarithmic factors. Toy experiments on a multivariate Normal distribution are provided to support the theoretical findings.

**Claims And Evidence:**

The paper is a theory paper, where all its theoretical claims have been rigorously proved under the stated assumptions. The authors do not overstate their contributions and clearly acknowledge the limitations of the work (e.g., the restrictiveness of the assumptions). Moreover, the toy experiments in Section 6 are consistent with and provide empirical support for the theoretical claims established in the previous sections.

**Essential References Not Discussed:**

As far as I know, the essential references have been discussed.

**Experimental Designs Or Analyses:**

As far as I can tell, the experiment section presents a toy multivariate Gaussian setting with the sole purpose of supporting the theoretical findings. The code is provided, but I did not verify it directly. However, the experimental results appear sound and provide appropriate empirical support for the theoretical claims.

**Methods And Evaluation Criteria:**

The paper primarily contributes to the theoretical aspects of preference learning. The main methodology described is the deterministic preferences MLE (DP-MLE) presented in Section 4, which uses a 0-1 loss to constrain the set of feasible parameters based on the implicit assumption that reward models are well-specified. This approach makes sense if there are good reasons to believe the reward model is indeed well-specified. However, I do have some concerns about the assumptions used in the analysis, which I will outline in the following sections.

**Other Comments Or Suggestions:**

1. The definition of 0-1 loss for stochastic preferences in lines 230-231 is unclear. I suggest the authors clarify what this means exactly and explicitly state why minimizing such a loss is NP-hard.

2. Theorem 4.3 is presented as a corollary of the main Theorem 4.8, yet it appears in an earlier section. This may cause confusion for readers. I suggest the authors first state Theorem 4.8 and then present the corollary specifically for Normal/Laplace distributions to improve logical flow.

3. The proof of Theorem 4.8 could be easier to grasp if the authors provided some intuition behind the definition of $V_{\theta^\star, u}$ on line 302.

4. There appears to be a typo in lines 431-432.

**Other Strengths And Weaknesses:**

**+ Soundness, Novelty, and Technical Contributions**

All assumptions are concretely specified, and I find the technical contribution around achieving the accelerated rate of $\mathcal{O}(\frac{1}{n})$ both interesting and novel. While I have concerns about the restrictiveness of the assumptions, the authors demonstrate that both Normal and Laplace distributions satisfy these assumptions, which can be seen as a meaningful degree of practicality. All results are well-supported by rigorous proofs and tested by the toy experiments.

**- Quality of Presentation**

The presentation of the work could be significantly improved. The current writing, especially in Sections 1 to 3, reads more like a collection of independent chunks of information without a cohesive story connecting them. I believe the authors have developed several ideas and attempted to articulate them, but in doing so, they relied on implicit contextual understanding that isn't provided in the text. I would suggest approaching the writing from the perspective of a reader encountering the paper with no prior knowledge of the work and providing sufficient context throughout. Additionally, in Section 2, definitions and motivations are sometimes intermingled, making it unclear what constitutes a formal definition versus what serves as intuitive examples. The following concrete instances illustrate these issues, though they are more cases:

1. I needed to read the entire paper first to understand the paragraph in lines 57-63 about how hard constraints can help achieve better estimation error for a standard normal distribution. The presentation would benefit from more context regarding what is known and not known by the estimator about the estimation task, what the parameter of interest actually represents, and why $S_i$ is defined in this way.

2. In Section 2, the paragraph about *informative preferences* (lines 137-150) defines the two sets $\mathcal{G}_0$ and $\mathcal{G}_1$ based on a vague notion of "informativeness" without providing context for why one might be interested in samples with non-zero preference gradients. What does it mean to say "only preferences of samples in $\mathcal{G}_1(\theta^\star)$ can provide information on $\theta^\star$"? Why is $\mathcal{G}_0$ defined if informativeness is only based on $\mathcal{G}_1$?

3. Also in Section 2, the paragraph on *negative examples* (lines 152-164) is extremely unclear. This paragraph could be placed anywhere in the paper without affecting the overall narrative. The claims lack concreteness, and no proof or proof sketch is provided.

**- Significance of the Results for the Community**

The main limitation of this work is perhaps its overly restrictive set of assumptions, which may limit its significance and applicability in the broader community, particularly in preference learning. Reading the first lines of the abstract, It seems that the authors motivate the applicability of their theory based on advances in preference learning for language models. While the authors acknowledge the restrictiveness of their assumptions, they rely on their results showing that Normal and Laplace distributions satisfy these assumptions to claim broader applicability. However, I do not believe that merely demonstrating compliance with these assumptions for Normal/Laplace distributions is sufficient to establish applicability in more complex scenarios like preference learning in language models. My concerns are twofold:

1. The deterministic method (DP-MLE) with 0-1 loss in Section 4 relies on the well-specification of the reward (preference) model. Although standard asymptotic theory for M-estimators also assumes well-specification of the parametric model class, there seems to be a big difference. In standard MLE, if the model is misspecified, one can employ quasi-MLE to obtain robust estimation (see [1]). However, in the deterministic case, if the reward model is misspecified, the constraint set $\mathcal{C}_n$ may not necessarily converge to a set containing the true parameter $\theta^\star$ as $n \to \infty$. Since DP-MLE is constrained to $\mathcal{C}_n$, I suspect it could yield arbitrary estimates under misspecified models and lack robustness. This is particularly concerning given that model misspecification is almost always a possibility, especially when dealing with human annotators, where reward models are known to be misspecified [2].

2. Even assuming correctly specified models, the paper provides no recipe to verify whether Assumptions 4.4, 4.5, and 4.7 hold for a given parametric model. These assumptions appear extremely difficult to check for arbitrary parametric models. The authors devote six pages of mathematical derivations just to prove them for the relatively simple cases of Normal and Laplace distributions.


**References**

[1] White, Halbert. "Maximum likelihood estimation of misspecified models." Econometrica: Journal of the econometric society (1982).

[2] Casper, Stephen, Xander Davies, Claudia Shi, Thomas Krendl Gilbert, Jérémy Scheurer, Javier Rando, Rachel Freedman et al. "Open problems and fundamental limitations of reinforcement learning from human feedback." arXiv preprint arXiv:2307.15217 (2023)

**Questions For Authors:**

Despite the limitations I mentioned, I still think the paper has the potential to be accepted at the conference. The deciding factor for me is the authors' response to the limitations I highlighted under "Significance of the Results for the Community" in the weaknesses section. Could the authors elaborate on the implications of model misspecification in DP-MLE and also explain how one can verify assumptions 4.4, 4.5, and 4.7 for realistic models? I don't necessarily expect proof that DP-MLE is robust to misspecification, but I would expect at minimum an acknowledgment of this as a major limitation of the work.

*Minor Questions:*

1. In Figure 1.a, how does the RU method outperform DP? This seems somewhat counter-intuitive. Could you elaborate on this observation?

2. In Section 6, what is the goal and implication of including the paragraph about covariance gap starting in line 434? It seems disconnected from the other points in the experiment section. Could you provide more context for its relevance?

3. How do you envision applying the DP-MLE method for preference learning in realistic language model training? Can you provide at least an outline of when/how this approach might be feasible in practice?

**Relation To Broader Scientific Literature:**

The contributions are directly related to the empirical success of preference-based fine-tuning of large language models through methods like RLHF, compared to methods that only rely on positive examples such as supervised fine-tuning. In this context, the work attempts to develop a deeper theoretical understanding of when such preferences can help improve learning, using a simplified parametric setting.

The ideas presented in Section 3, regarding the effect of adding preferences to standard M-estimators, are primarily based on the well-established asymptotic normality theory of parametric MLE. The authors apply similar techniques and tools to calculate the Fisher information matrix in the preference-based setup and investigate the conditions under which it can be strictly more informative than the standard M-estimator.

The results in Section 4, however, appear more novel and rely on the hard constraints imposed by deterministic preferences. The authors have appropriately discussed previous related results that use similar hard constraints to achieve better estimation error: for example, the parameter estimation of a uniform distribution on $[\theta, \theta + 1]$ by taking the minimum of samples (Wainwright, 2019), which has a known minimax rate of $\Theta(\frac{1}{n})$.

**Theoretical Claims:**

The paper has several theoretical claims:
1. ${\color{green}\text{Lemma 3.1}}$ and Lemma 3.2 on the asymptotic variance of the preference-based M-estimators.
2. ${\color{green}\text{Lemma 4.1}}$ on the benefits of using their proposed constraint-based estimator compared to M-estimators for Normal distributions.
3. ${\color{green}\text{Theorem 4.8}}$ (including ${\color{green}\text{Lemma 4.6}}$) and its corollary ${\color{green}\text{Theorem 4.3}}$ on proving the accelerated rate of $\mathcal{O}(\frac{1}{n})$ for their proposed estimator.
4. Theorem 5.3 (including Lemma 5.1) on the estimation lower bound for the deterministic feedback case.
5. Proving that the Normal and Laplace distributions satisfy Assumptions 4.2, 4.4, 4.5, 4.7, and 5.2, which are necessary for the theoretical claims in the paper (Appendices E, F).

I have only checked the correctness of the results shown in ${\color{green}\text{green}}$ and did not find any issues.

---

> ### Author Rebuttal · Authors · 2025-03-28
>
> We thank Reviewer vgrP for the time spent and the detailed comments. Due to the limited space, we only address some of the reviewer’s concerns.
>
> **Restrictive assumptions**
> While our research question is inspired by iterative human preference alignment (see the answer to Reviewer Gth4), we do not claim the direct applicability of DP MLE for realistic LLM training. When studying DP MLE only, we conjecture that the “global” assumptions 4.2 and 4.4 can be weakened to local versions. Using time-uniform concentration results, we can build a sequence of shrinking confidence regions $(R_n)_n$ around SO MLE that contains $\theta^\star$ for all time $n$ with high probability (whp). Then, we modify DP MLE to be constrained on $R_n \cap C_n$ that contains $\theta^\star$ whp. For $n$ large enough, $R_n \cap C_n$ will be included in a local neighborhood of $\theta^\star$ under which the “local” assumptions 4.2 and 4.4 are satisfied. Given that Assumption 4.4 is based on “ignoring” the reminder term in a first-order Taylor expansion, assuming a local version is a significantly weaker requirement.
>
> **1. Misspecification**
> There are two possible sources of misspecification not taken into account by our current analysis.
> - Misspecified observations. See answer to Reviewer yquN when $\theta^\star \notin \Theta$. When $p^\star \notin F$, $L_n(\theta)$ is a quasi-log-likelihood term, as $F$ doesn’t contain the true structure. Under sufficient regularity, SO quasi-MLE converges towards $\theta_{0} \in argmin_{\theta \in \Theta} KL(p^{\star}, p_{\theta})$ where $ p^\star \ne p_{\theta_0} \in F$. Without the separability from well-specified deterministic preference, we define DP quasi-MLE as in the answer to Reviewer yquN. Under sufficient regularity, this estimator converges towards the minimizer of a similar optimization problem based on the above KL and a misspecified equivalent of $m(\theta)$.
> - Misspecified preferences. The Bradley-Terry (BT) model that uses reward-based preferences has limited expressivity as it doesn’t allow for intransitive preferences. Even when individuals exhibit transitive preferences, their averaged preferences might be intransitive due to disagreements. See Munos et al. (2024, Nash Learning from Human Feedback) or Swamy et al. (2024, A Minimaximalist Approach to Reinforcement Learning from Human Feedback).
>
> **2. Verifying our assumptions**
> In all generality, it is challenging to give a general recipe to formally verify those assumptions. A formal verifier (Lean) or software (SageMath) might be useful given a closed-form definition. Numerically, those assumptions can be confirmed or rejected by sampling from $p_{\theta^\star}^{\otimes 2}$. Assumption 4.4 is rejected by exhibiting $(X_i,Y_i) \in \tilde D(\theta^\star,\theta) \setminus D(\theta^\star,\theta)$. Assumptions 4.2 and 4.5 are confirmed by finding $(X_i,Y_i) \in D(\theta^\star,\theta)$ and $(X_i,Y_i) \in G_{1}(\theta^\star,u)$. The sampling complexity of such tests scales as the inverse event’s probability. Using Dvoretzky–Kiefer–Wolfowitz inequality, $F_{\theta^\star,u}$ can be estimated to verify Assumption 4.7 hold.
> Our additional experiments with accelerated rate include Laplace and Rayleigh distributions, see the anonymous plots at https://anonymous.4open.science/r/ICML25SuppExp .
>
> **RU versus DP**
> See answer to Reviewer Gth4.
>
> **Covariance Gap**
> Our simulations suggest that the asymptotic gaps between SO and SP are mild. The empirical gap is also small for moderate $n$.
>
> **0-1 loss**
> See answer to Reviewer yquN. For non-separable data, the minimization of the 0-1 classification loss can be NP-hard even for the simple class of linear classifiers, e.g., Feldman et al. (2018, Agnostic Learning of Monomials by Halfspaces is Hard).
> Inspired by (Tang et al., 2024, Generalized Preference Optimization: A Unified Approach to Offline Alignment), we implement estimators based on other convex surrogates: Hinge, Square, Truncated square, Savage, and Exponential. All estimators perform on par with the logistic loss, see the plot at https://anonymous.4open.science/r/ICML25SuppExp .
>
> **Intuition on $V_{\theta^\star,u}$**
> It quantifies the amount of information in $(X_i,Y_i)$ to discriminate $\theta^\star$ from other parameters on the half-line directed by $u$. The lower $V_{\theta^\star,u}(X_i,Y_i)$ is, the more discriminative $(X_i,Y_i)$ is.
>
> **Informative preferences**
> For observations with null preference gradient, parameters close to $\theta^\star$ could have similar preferences. Therefore, those samples are not sufficient to discriminate between them.
>
> **Negative examples**
> Those claims are a direct consequence of the definitions and will be proved in Appendix for completeness.
>
> **Typo**
> It will be fixed.

---

> > ### Comment · Reviewer_vgrP · 2025-04-02
> >
> > Thanks for clarifying the main concerns. I’ve raised my score based on the additional context you provided. However, I would still like this discussion—at least in part—to be included in the main paper, especially the sections on restrictive assumptions and misspecification. For this reason, I vote for acceptance, conditional on updating the camera-ready paper based on this discussion.

---

> > > ### Author Response · Authors · 2025-04-03
> > >
> > > We appreciate the reviewer’s support for the acceptance of our work based on the additional context. We will use the extra page in the main paper to include these interesting discussions, such as misspecification, verification and relaxation of our assumptions, alternative reward models, and more detailed intuitions. Additionally, we will expand the Appendices with those supplementary experiments and provide detailed proofs to support our added comments.

---

### Decision · Program_Chairs · 2025-05-01

**Decision:**

Accept (spotlight poster)

**Comment:**

The paper studies shws how learning with preferences can improve parameter estimation rates for Gaussian and Laplace distributions. They show that for estimators based on hard preferences the error converges at a rate of O(1/n) and that this rate is mini-max optimal up to dimension and problem-dependent constants. The reviewers found the paper measured in its statements, the contributions novel, and well supported by theory and experiments. All reviewers agreed the work should be accepted. Given the widespread use of supervised fine-tuning and preference based learning to align LLMs, I do think some of the insights here can be translated into practical learning algorithms to improve LLMs -- this may be worth commenting on within the discussion in terms of the practical ramifications of this work. I recommend the authors use the extra page to incorporate vgrP's suggestions to reframe the writing for sections 1-3 with an emphasis on clarity since doing so should mitigate some of the questions raised by other reviewers as well.